# Structure and topology around the cleavage site regulate post-translational cleavage of the HIV-1 gp160 signal peptide

Erik Lee Snapp[1†], Nicholas McCaul[2†], Matthias Quandte[2‡], Zuzana Cabartova[3], Ilja Bontjer[4], Carolina Källgren[5,6], IngMarie Nilsson[5,6], Aafke Land[2§], Gunnar von Heijne[5,6], Rogier W Sanders[4], Ineke Braakman[2]*

[1]Janelia Research Campus, Howard Hughes Medical Institute, Ashburn, United States; [2]Cellular Protein Chemistry, Bijvoet Center for Biomolecular Research, Faculty of Science, Utrecht University, Utrecht, Netherlands; [3]National Institute of Public Health, National Reference Laboratory for Viral Hepatitis, Prague, Czech Republic; [4]Department of Medical Microbiology, Laboratory of Experimental Virology, Center for Infection and Immunity Amsterdam, Academic Medical Center, Amsterdam, Netherlands; [5]Department of Biochemistry and Biophysics, Stockholm University, Stockholm, Sweden; [6]Science for Life Laboratory, Stockholm University, Solna, Sweden

*For correspondence:
i.braakman@uu.nl

[†]These authors contributed equally to this work

Present address: [‡]dr heinekamp Benelux BV, Riethoven, Netherlands; [§]Hogeschool Utrecht, Institute of Life Sciences, Utrecht, Netherlands

**Abstract** Like all other secretory proteins, the HIV-1 envelope glycoprotein gp160 is targeted to the endoplasmic reticulum (ER) by its signal peptide during synthesis. Proper gp160 folding in the ER requires core glycosylation, disulfide-bond formation and proline isomerization. Signal-peptide cleavage occurs only late after gp160 chain termination and is dependent on folding of the soluble subunit gp120 to a near-native conformation. We here detail the mechanism by which co-translational signal-peptide cleavage is prevented. Conserved residues from the signal peptide and residues downstream of the canonical cleavage site form an extended alpha-helix in the ER membrane, which covers the cleavage site, thus preventing cleavage. A point mutation in the signal peptide breaks the alpha helix allowing co-translational cleavage. We demonstrate that postponed cleavage of gp160 enhances functional folding of the molecule. The change to early cleavage results in decreased viral fitness compared to wild-type HIV.
DOI: https://doi.org/10.7554/eLife.26067.001

## Introduction

Proteins destined for the secretory pathway are translated and translocated into the endoplasmic reticulum (ER), which provides a specialized environment for their folding, disulfide bond formation, and N-linked glycosylation. Targeting of soluble and type-I transmembrane proteins to the ER is mediated via cleavable signal peptides, near-N-terminal hydrophobic stretches of 14–50 amino acids that are recognized by SRP (*von Heijne, 1985*; *Kurzchalia et al., 1986*; *Lütcke et al., 1992*; *Walter and Blobel, 1981*; *Blobel and Dobberstein, 1975*; *Hegde and Bernstein, 2006*). Cleavable signal peptides are variable in sequence but share characteristics of an N-terminal region with typically 0–2 basic residues, a membrane-spanning hydrophobic α-helix (H) region, and a C-terminal region that often contains a signal-peptide cleavage site (*von Heijne, 1983*; *von Heijne, 1984*).

Signal-peptide cleavage is mediated by the signal-peptidase complex, which, like oligosaccharyl transferase, associates with the translocon (*Görlich et al., 1992*; *Gilmore, 1993*). In a second

cleavage step signal peptides are cleared from the ER membrane by signal-peptide peptidase, an intramembrane rhomboid-like protease (*Weihofen et al., 2002*). Signal peptides are widely believed to be cleaved co-translationally (*Blobel and Dobberstein, 1975*; *Jackson and Blobel, 1977*; *Martoglio and Dobberstein, 1998*), but cleavage may well occur anywhere from early co-translational to late post-translational, depending on the protein. The minimal requirement for cleavage is the emergence of the cleavage site in the ER lumen, which translates to ~70 synthesized residues of a nascent polypeptide chain (*Daniels et al., 2003*; *Kowarik et al., 2002*; *Hou et al., 2012*), and its recognition by the signal-peptidase complex. Yet, examples such as preprolactin, influenza virus hemagglutinin (HA) and human cytomegalovirus US11 show that cleavage can follow initial folding and/or glycosylation and sometimes requires longer nascent-chain lengths (*Daniels et al., 2003*; *Rutkowski et al., 2003*; *Rehm et al., 2001*).

The HIV-1 envelope glycoprotein gp160 represents an extreme case of late post-translational cleavage and the mechanism for delayed cleavage has been a long-standing problem (*Li et al., 2000*; *Land et al., 2003*; *Li et al., 1996*). The gp160 signal peptide acts as a membrane tether for at least 15 min after synthesis and requires at least some folding of the ectodomain for cleavage (*Land et al., 2003*). When proper folding is prevented by maintaining gp160 in a reduced state with DTT treatment or by blocking N-linked glycosylation using tunicamycin, the signal remains uncleaved and gp160 fails to exit the ER (*Land et al., 2003*). Gp160 begins as a transmembrane protein whose folding requires formation of conserved disulfide bonds, abundant glycosylation, proline isomerization, and trimerization before the protein leaves the ER (*Land et al., 2003*; *Willey et al., 1988*; *Earl et al., 1990*; *Earl et al., 1991*; *Land and Braakman, 2001*; *van Anken et al., 2008*). In the Golgi complex, gp160 glycans are modified and furin proteases cleave the trimeric glycoprotein into its two subunits, which stay non-covalently attached and are incorporated into new virions at the plasma membrane (*Araújo and Almeida, 2013*; *Sundquist and Kräusslich, 2012*). Gp160 is essential for virus entry as gp120 recognizes CD4, the HIV receptor on the cell surface, and one of the HIV coreceptors CCR5 and CXCR4, whereas gp41 mediates fusion with the target membrane (*Blumenthal et al., 2012*). The soluble subunit gp120 dominates gp160 folding and can fold and be secreted on its own, in the absence of gp41 (*Land et al., 2003*). Signal-peptide cleavage of gp120 is similarly delayed as for full-length gp160 and obeys the same rules (*Land et al., 2003*; *van Anken et al., 2008*). Therefore, for certain assays, we use gp120 interchangeably with gp160.

Late cleavage of a signal peptide requires a two-component mechanism that prevents initial recognition of the cleavage site by the signal-peptidase complex and then enables cleavage at a later time. In this study, we identify the structural basis of delayed signal-peptide cleavage and propose a novel role for gp160 signal-peptide cleavage as a built-in quality control mechanism.

## Results

### Residues up- and downstream of the signal-peptide cleavage site prevent co-translational cleavage

To examine whether the signal peptide directly impaired co-translational cleavage, we replaced the natural gp160 signal peptide (30 residues) with the unrelated signal peptides of influenza virus hemagglutinin (HA), cystatin (cys), Igκ, and two synthetic, optimal (opt), or suboptimal (sub) signal peptides (*Barash et al., 2002*). The timing of signal-peptide cleavage was assayed using a radioactive pulse-labeling approach (*Land et al., 2003*). Briefly, HeLa cells expressing wild-type or mutant gp120 were incubated with $^{35}$S-methionine and $^{35}$S-cysteine for 5 min and chased for the indicated times. Cells were transferred to 4°C and iodoacetamide was added to prevent further formation and isomerization of disulfide bonds. Cells were lysed in Triton X-100 and the detergent lysates were immunoprecipitated with the polyclonal antibody 40336, which recognizes all forms of gp160. The immunoprecipitates were deglycosylated and subjected to reducing 7.5% SDS-PAGE to resolve signal-peptide-cleaved and uncleaved forms.

After completion of synthesis, unprocessed deglycosylated gp120, with its signal peptide still attached (Ru) runs as a single band of ~70 kDa (*Land et al., 2003*) (*Figure 1A*). After a 60-min chase, a second band with increased mobility appears, representing signal-peptide-cleaved gp120 (Rc). In contrast, immediately after synthesis all gp120 constructs with non-native signal peptides already

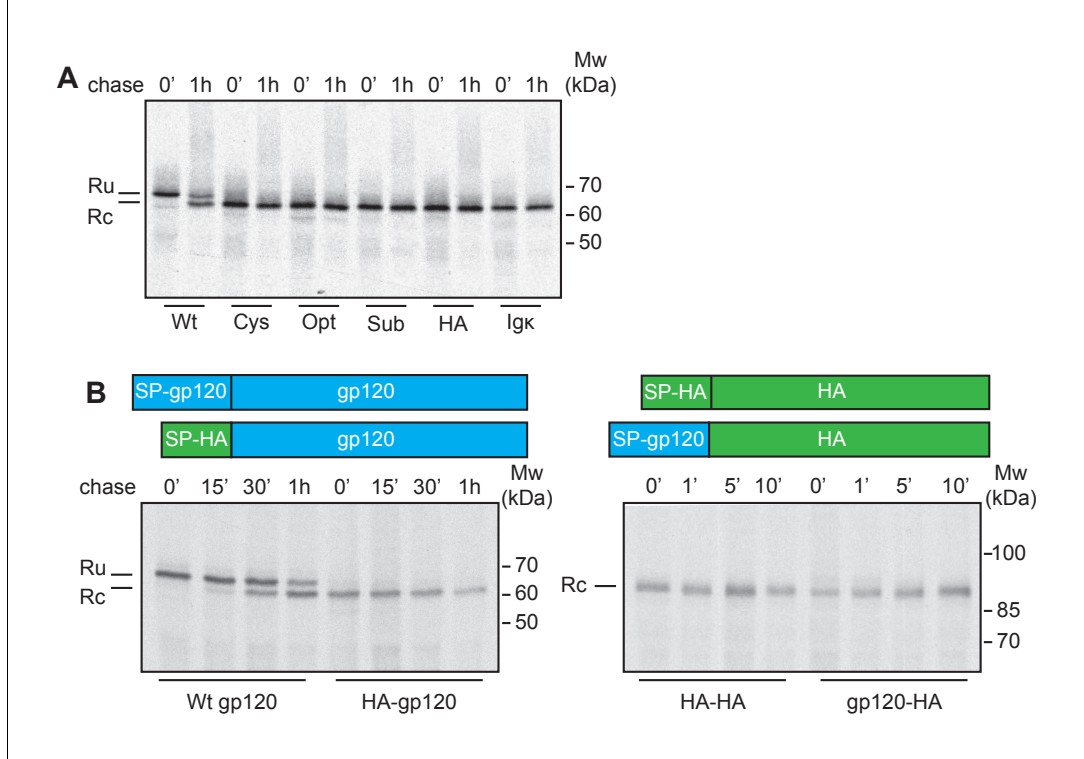

**Figure 1.** The interplay of gp120 and its natural signal peptide causes post-translational cleavage. (**A**) HeLa cells expressing wild-type gp120 (Wt) and gp120 with exogenous signal peptides, HA (HA), Ig κ (Ig κ), cystatin (Cys), Optimal (Opt), and Suboptimal (Sub), were radiolabeled for 10 min and chased for 1 hr or not (0'). After immunoprecipitation gp120 samples were deglycosylated and analyzed using reducing 7.5% SDS-PAGE. Gels were dried and exposed to Kodak-MR films. (**B**) As in (**A**) except Wt gp120, gp120 with the signal peptide of HA (HA-gp120), wild-type HA (HA-HA) and HA with the signal peptide of gp120 (gp120-HA) were labeled for 5 min and chased for indicated times. Ru: reduced gp120 with the signal peptide still attached; Rc: signal-peptide cleaved gp120; wt: wild-type gp120. Gels shown are representative of at least three independent experiments performed with fresh cells and transfections (biological replicates).

DOI: https://doi.org/10.7554/eLife.26067.002

ran as a single band at the position of signal-peptide-cleaved gp120, indicating that the gp120 signal peptide is necessary for delayed signal cleavage.

We next investigated whether the gp120 signal, alone, was sufficient for post-translational cleavage, and swapped the signal peptides of gp120 and the unrelated transmembrane protein, influenza virus HA. The HA-signal-peptide construct (HA-gp120) again was detected as co-translationally cleaved gp120 (*Figure 1B*). The gp120 signal peptide alone hence did not convert HA into a post-translationally cleaved protein. Both wild-type and signal-peptide-swapped HA (gp120-HA) ran as a single band on the gel. Therefore, features of both mature gp120 and its signal peptide are necessary for post-translational cleavage.

To establish the minimal requirements of the gp160 sequence sufficient for post-translational cleavage, we designed a series of GFP constructs that contained the gp160 signal peptide and an increasing number of mature gp160 residues before GFP (*Figure 2A*, *Figure 2—figure supplement 1A*). GFP expression and localization were monitored by fluorescence microscopy (*Figure 2—figure supplement 1B*) and signal-peptide cleavage was assessed by immunoblot (*Figure 2B*).

As expected, GFP attached to the gp160 signal peptide was correctly targeted to the ER and accumulated in the ER due to the engineered retrieval signal KDEL (*Figure 2—figure supplement 1B* SP+1, SP+10). Immunoblot analysis revealed a single band of GFP when one residue of mature gp160 sequence linked the signal peptide and GFP (SP+1), which migrated at the same position as control, GFP-KDEL (*Figure 2B*). Addition of four more residues of mature gp160 (SP+5) delayed the processing and resulted in signal-cleaved and uncleaved GFP species. Processing decreased even more with additional gp160 residues and the inclusion of ~10 downstream residues turned the signal

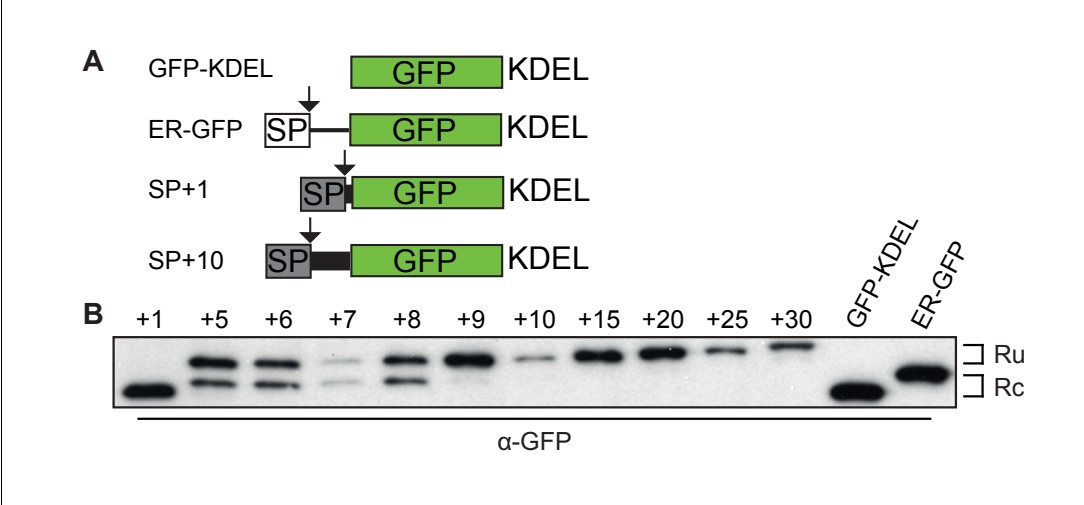

**Figure 2.** Downstream residues of gp160 regulate signal-peptide cleavage. (**A**) Schematic of different GFP reporter constructs generated for imaging and immunoblot experiments. All reporters have the identical monomeric GFP-KDEL cassette. For ER GFP, the GFP-KDEL cassette is preceded by the bovine prolactin signal peptide (open box SP). The reporters with the HIV Env signal peptide (grey box SP) are fused to one or more amino acids of the mature gp120 domain followed by the GFP-KDEL cassette. (**B**) Western Blot analysis of control constructs (GFP-KDEL, ER-GFP) or gp160 signal peptide with 1-30 downstream residues (SP +1, . . ., 30). GFP-KDEL and SP +1 run lower on gel as they lack the additional residues downstream of the cleavage site. Ru: unprocessed molecules with signal peptide still attached; Rc: signal peptide-cleaved molecules. All images shown are representative of at least two independent experiments performed with fresh cells and transfections (biological replicates).

DOI: https://doi.org/10.7554/eLife.26067.003

The following figure supplement is available for figure 2:

**Figure supplement 1.** Characterization of SP-GFP fusion constructs.

DOI: https://doi.org/10.7554/eLife.26067.004

peptide into a predominantly uncleaved signal anchor. We examined whether the population of cleaved and uncleaved molecules in steady state represented distinct populations or a low rate of cleavage. Pulse-chase experiments showed that cleavage only occurred during the pulse-labeling period and not anymore after synthesis (*Figure 2—figure supplement 1C and D*), demonstrating that signal-peptide cleavage of the GFP-reporter constructs was not progressive. We concluded that gp160 residues downstream of the signal cleavage site, in combination with the signal peptide itself, were sufficient to modulate cleavage efficiency of the gp160 signal peptide.

## Post-translational cleavage is conserved across HIV-1 subtypes

Studies to date have primarily focused on the efficiency of signal-peptide cleavage of two closely-related isolates of HIV-1 Env, HXB2 (*Li et al., 2000*) and LAI (*Land et al., 2003*), which differ by only a single amino acid in the signal peptide (M24 in HXB2 and I24 in LAI) and 25 residues in gp120. To establish whether post-translational signal-peptide cleavage is conserved between different HIV-1 subtypes we examined the consensus sequence for each subtype and compared them to the sequence of subtype B strain HXB2 (*Figure 3A*). The amino acids responsible for the switch from signal peptide to signal anchor (*Figure 2AB*, +4 to +10) were shown to be highly conserved across all subtypes. Indeed, an alignment of more than 4100 sequences shows 99–100% conservation of these amino acids, with the exception of L34 which was 89% conserved (*Supplementary file 1*). As the residues immediately before and after the cleavage site are involved in recognition by the signal-peptidase complex (*von Heijne, 1983*; *Choo and Ranganathan, 2008*), we also examined the residues from positions −5 to +3 relative to the cleavage site. While the precise amino-acid conservation varied (16% for K33% to 87% for C28), amino-acid differences between sequences largely conserved the character of the amino acid in question (*Supplementary file 1*).

The subtype-C consensus sequence demonstrated the greatest deviation from the HXB2 sequence in and around the cleavage site (*Figure 3A*). We therefore chose a subtype-C isolate with

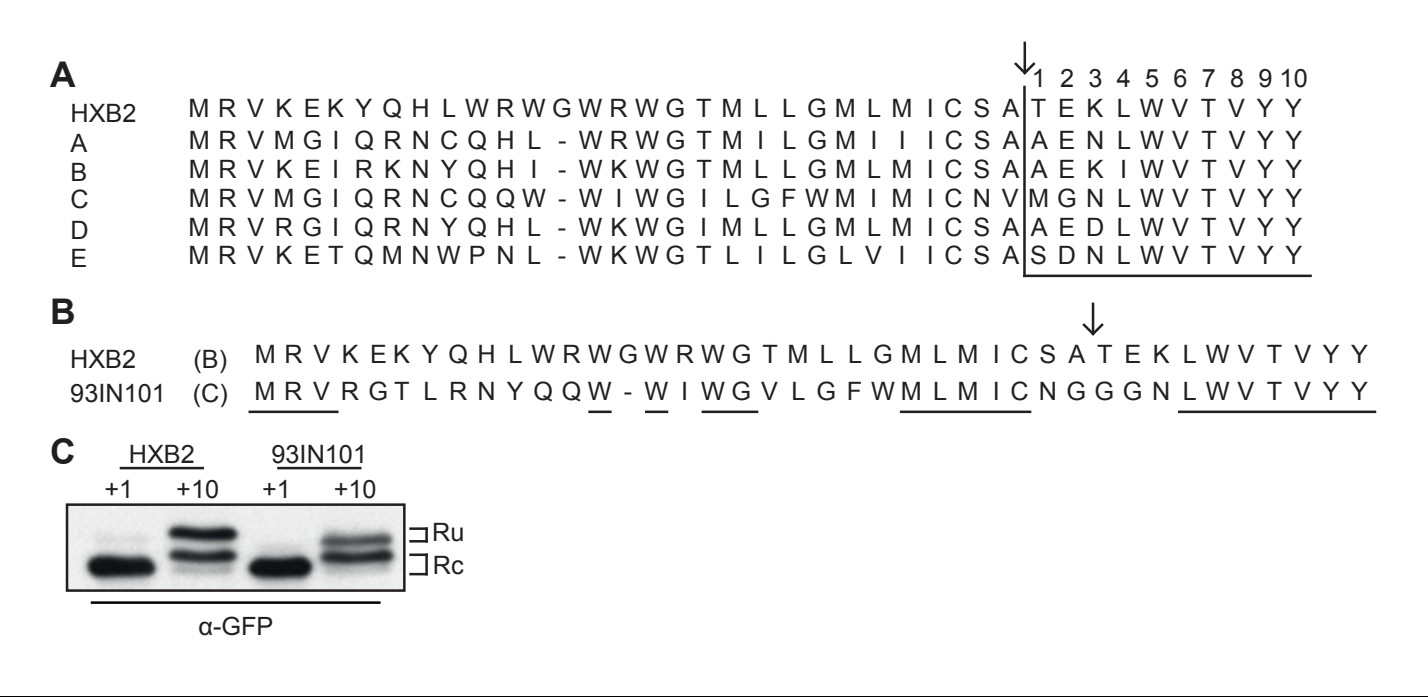

**Figure 3.** Env posttranslational signal-peptide cleavage is conserved between subtypes. (**A**) Alignment of the signal peptide of HIV-1 Env reference strain HXB2 and the consensus sequences for subtypes A-E (www.hiv.lanl.gov). (**B**) Sequence alignment of HXB2 and subtype C strain 93IN101. Residues underlined are conserved between the two strains. (**C**) Western Blot analysis of HXB2 and 93IN101 SP +1 GFP and SP +10 GFP reporters. Blot shown is representative of two independent experiments performed with fresh cells and transfections (biological replicates).
DOI: https://doi.org/10.7554/eLife.26067.005

poor sequence conservation (93IN101), to compare signal-peptide cleavage with HXB2, using the GFP reporter system. Despite the low sequence similarity between the two isolates (*Figure 3B*), results for both 93IN101 SP+1 and SP+10 were comparable to those of HXB2, with efficient processing of SP+1 and delayed processing of SP+10. Notably the C isolate had a different ratio of cleaved and uncleaved, consistent with its slightly more efficient processing. Nevertheless, we concluded that post-translational signal-peptide cleavage is a conserved phenomenon across HIV-1 subtypes.

## The signal peptide acts as a signal anchor before cleavage

We asked where the uncleaved signal peptide resides for the unprocessed protein. We hypothesized that the uncleaved signal peptide was either trapped in the Sec61 translocon, where signal peptidase is thought to reside primarily (*Gilmore, 1993*) or that the uncleaved protein was released laterally into the ER membrane where the uncleaved signal peptide may act as a transient signal anchor.

To distinguish between these possibilities, we performed two different complementary assays. First, we assessed the mobility of several constructs in live cells using photobleaching analyses. A signal anchor should behave like a single-pass transmembrane protein, with a characteristic diffusion coefficient, while a protein trapped in the Sec61 translocon should exhibit very low mobility, as has been reported for the translocon (*Nikonov et al., 2002*). We compared diffusion rates of different GFP constructs determined from fluorescence recovery after photobleaching (FRAP) measurements (*Figure 4A and B*). Diffusion coefficients (*D*) differ from fast soluble proteins in the ER lumen ($D = 8–12 \ \mu m/s^2$), to slower transmembrane proteins ($D = 0.3–1 \ \mu m/s^2$), and very slow proteins attached to the translocon/ribosome complex ($D = 0.04 \ \mu m/s^2$) (*Nikonov et al., 2002*). Signal-peptide-cleaved GFP (*Figure 4B*, SP+1) diffused rapidly ($D = 9.6 \pm 2.7 \ \mu m/s^2$), comparable with the ER-GFP control ($D = 10.2 \pm 1.6 \ \mu m/s^2$) and indicative of a soluble protein. Addition of ten gp160 residues downstream of the cleavage site to GFP (SP+10) slowed signal-peptide cleavage dramatically and resulted in a much lower $D$ ($D = 1.7 \pm 0.6 \ \mu m/s^2$). The value was too high for a membrane protein, much too

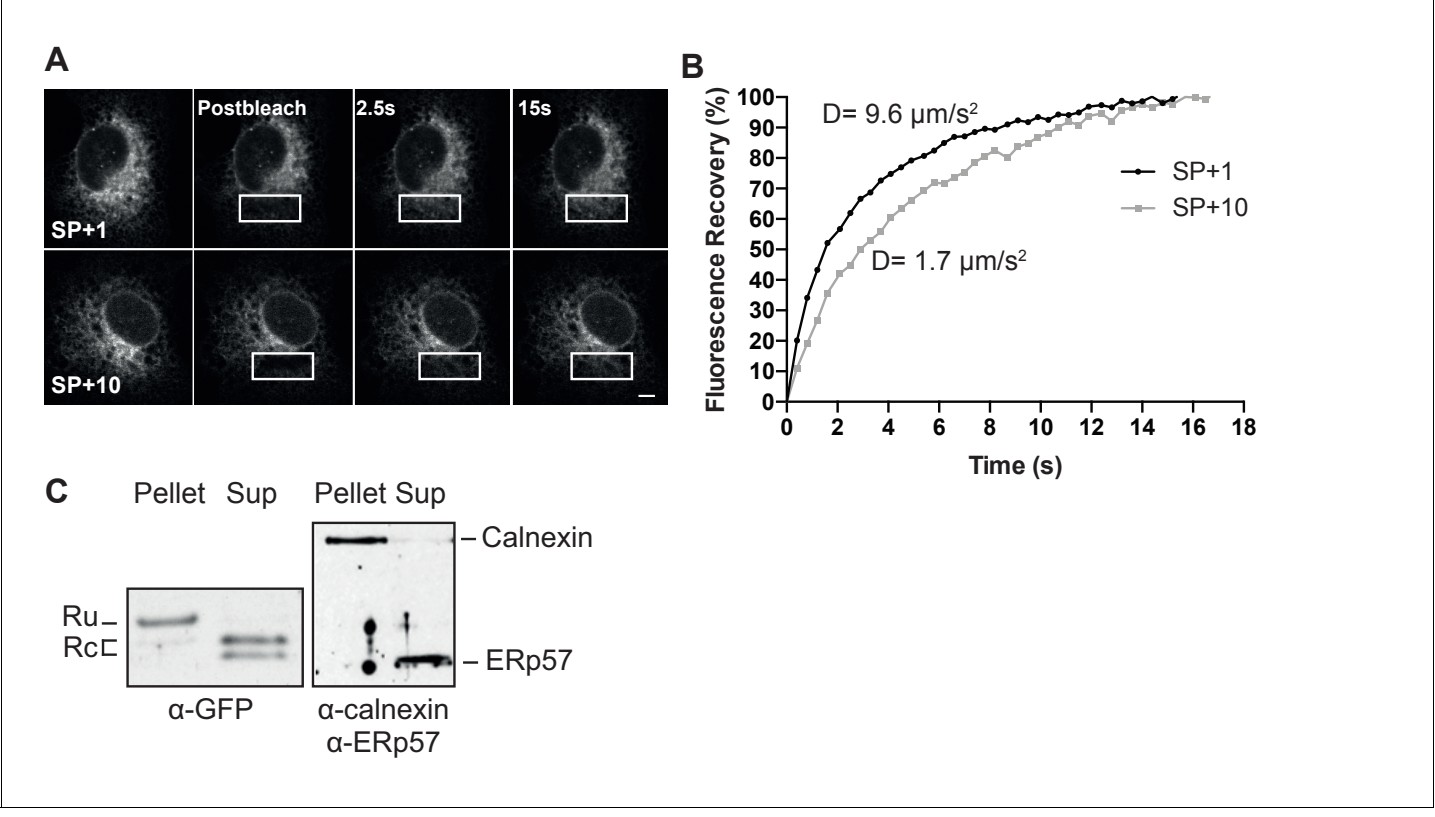

**Figure 4.** The uncleaved signal peptide acts as a signal anchor. (**A**) FRAP analysis of gp160 signal-peptide constructs as in *Figure 2*. Cos-7 cells expressing SP+1 and SP+10 GFP reporters were subjected to FRAP analysis. A small region of interest (white outlined box) was photobleached with intense laser light and imaged with low laser light to visually (**A**) and quantitatively (**B**) compare mobilities and fluorescence-intensity recovery rates. (**A**) Both reporters are mobile and unbleached reporters diffuse into the photobleached region of interest. Scale bar = 10 µm. (**B**) Plot of representative fluorescence recoveries into the photobleach region of interest reveals that SP+10 is slower to recover. Number of cells analyzed, diffusion constants and, statistical values can be found in *Figure 4—source data 1*. (**C**) Western Blot analysis of pellet and supernatant (sup) fractions from a carbonate extraction of cells expressing SP+10. Split band for GFP in Sup is likely due to fragmentation (*Wei et al., 2015*). Blots in panel C are representative of at least two independent experiments (biological replicates).

DOI: https://doi.org/10.7554/eLife.26067.006

The following source data is available for figure 4:

**Source data 1.** Summary of FRAP data.

DOI: https://doi.org/10.7554/eLife.26067.007

fast for a translocon protein, but much slower than for a soluble protein. The immunoblot data suggested that a mixed population is present in cells and our data are consistent with membrane and soluble species contributing to the diffusion coefficient. To validate membrane localization and to rule out that uncleaved SP+10 GFP was retained in the ER through trapping in the translocon, we used a carbonate-extraction approach, which releases translocon-bound clients into solution (*Görlich et al., 1992*) and retain integral membrane proteins in the pellet fraction (*Figure 4C*). Signal-cleaved SP+10 was fully soluble, while uncleaved SP+10 GFP was only found in the pellet fraction, indicating that it was fully integrated in the membrane and not in the translocon protein channel. Thus, the presence of a minimum of 5 to 10 gp120 amino acids downstream of the cleavage site prevented co-translational signal-peptide cleavage an anchored GFP to the membrane through this signal peptide.

## The signal-peptide-cleavage site is at the membrane-lumen interface

To establish whether the signal-peptide-cleavage site is exposed outside the ER membrane in wild-type gp120, we determined which residue is the first protruding into the ER lumen by in-vitro translation of a construct composed of 88 N-terminal gp120 residues (including the signal peptide)

coupled to the P2 domain of leader peptidase (Lep) in the presence of rough microsomes as a source of ER membranes (*Nilsson and von Heijne, 1993*) (*Figure 5A*). The construct contains two glycosylation sites. The first, within Lep, was readily glycosylated and served as a translocation control. The second glycosylation site was moved through positions 39–47 to determine the position at which it became glycosylated. N-linked glycosylation requires a minimal distance from the ER membrane of ~11–12 residues and therefore, glycosylation can be used as a 'molecular ruler' of lumenal exposure (*Nilsson and von Heijne, 1993*; *Bañó-Polo et al., 2011*).

Positions 39 and 40 were poorly glycosylated and thus located close to the ER membrane (*Figure 5B and C*). Glycosylation of the second site sharply increased at position 41 and beyond, which thus is located 11–12 residues away from the membrane-water interface. This result strongly supports the interpretation that the cleavage site of gp160's signal peptide is at the membrane-

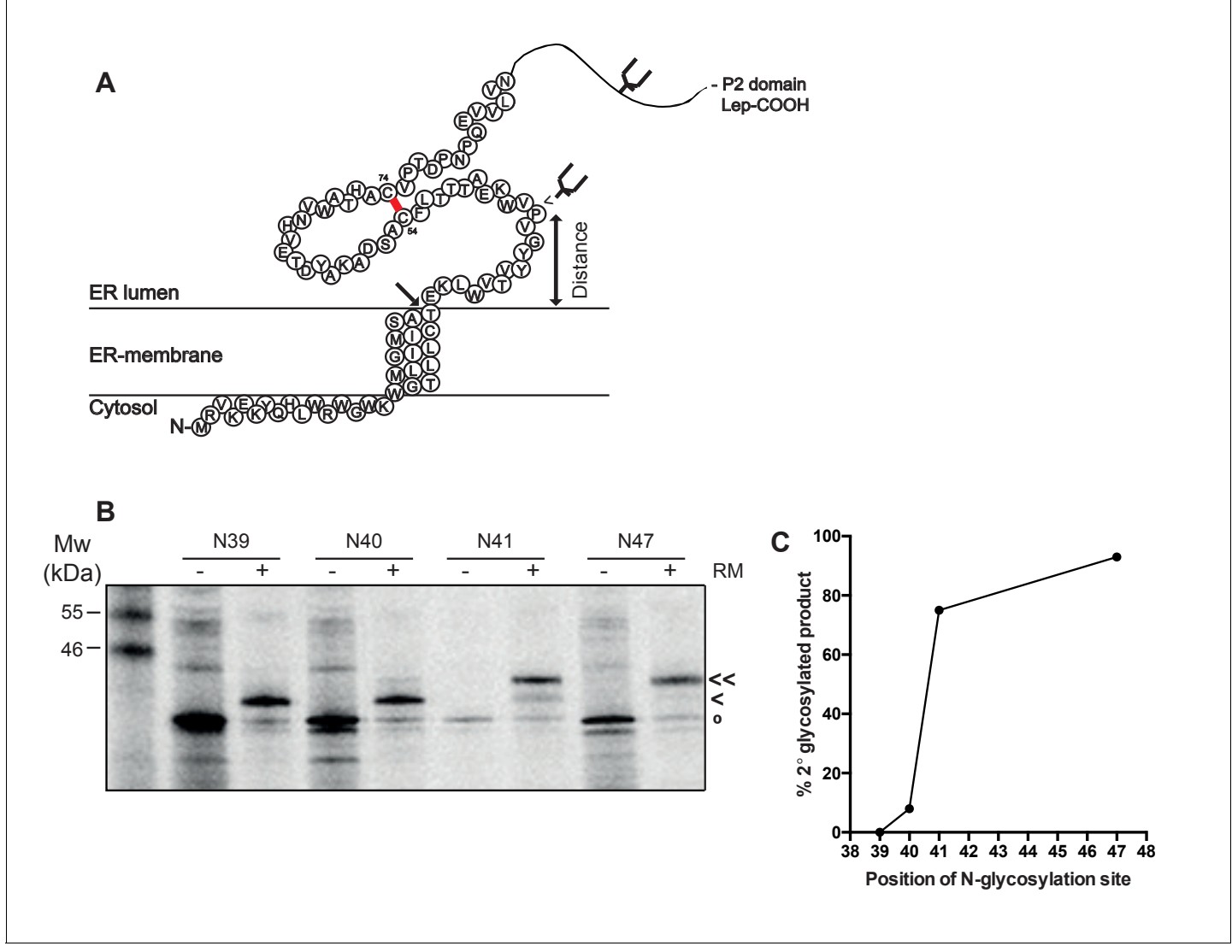

**Figure 5.** The signal-peptide cleavage site is buried in the membrane. (**A**) Cartoon of the first 89 residues of gp160 attached to the P2 domain of protein leader peptidase (Lep) at the C-terminus. A first glycosylation site in Lep was used as translocation control. The second site was introduced at positions 39–47 to determine at which position the distance to the membrane is sufficient for glycosylation. Arrow indicates site of signal-peptide cleavage. (**B**) Gp160-Lep constructs with glycosylation sites N39, N40, N41, and N47, were in-vitro translated in presence (+) or absence (-) of dog pancreas microsomes (RM). Gp160-Lep received a single (<) or double (<<) glycan, or remained unglycosylated/untranslocated (o). (**C**) Band intensities of B were quantified and percentage of second glycosylation product was plotted against position of the engineered glycosylation site. The data in panels B and C are representative of multiple independent experiments (biological replicates).
DOI: https://doi.org/10.7554/eLife.26067.008

lumen interface, with T31 barely exposed in the ER lumen. The active site of signal peptidase is predicted to be 0.4–1.1 nm beyond the membrane surface (*Liang et al., 2003*). It therefore is likely that the shielding of the cleavage site by membrane phospholipids contributes to the delayed cleavage of the signal peptide.

## Secondary structure prevents co-translational signal-peptide cleavage

Taken together, our results suggest a model in which the signal peptide and flanking gp120 sequence are sufficient to delay signal-peptide cleavage and that the cleavage site is poorly accessible. The signal peptide and flanking domain hence appear to regulate signal-peptide cleavage. To characterize the physical properties of the gp160 signal peptide, the sequence was analyzed with the online prediction tool SignalP (*Bendtsen et al., 2004*). A classical signal peptide, prolactin, exhibits a clear separation of N-, H-, and C-regions and a high signal-peptide probability in the hidden Markov model (*Bendtsen et al., 2004*) (*Figure 6A*). In contrast, gp160's signal peptide contains a predicted hydrophobic H-region that overlaps substantially with the C-region containing the cleavage site (*Figure 6A*). This predicted structure was highly unusual, as analyses of numerous other cleaved signal peptides did not exhibit overlap of H and C regions (our unpublished data). The probabilities for the gp160 signal peptide to be a cleavable signal peptide or an uncleaved signal anchor were 0.628 and 0.358, respectively. Similar probabilities were obtained for gp160 sequences from five different subtypes. SignalP analysis further corroborated our biochemical findings that the downstream residues of the mature gp120 influence probability of signal-peptide cleavage. When we tested the impact of the mature HA sequence on the gp160 signal peptide, separation between the

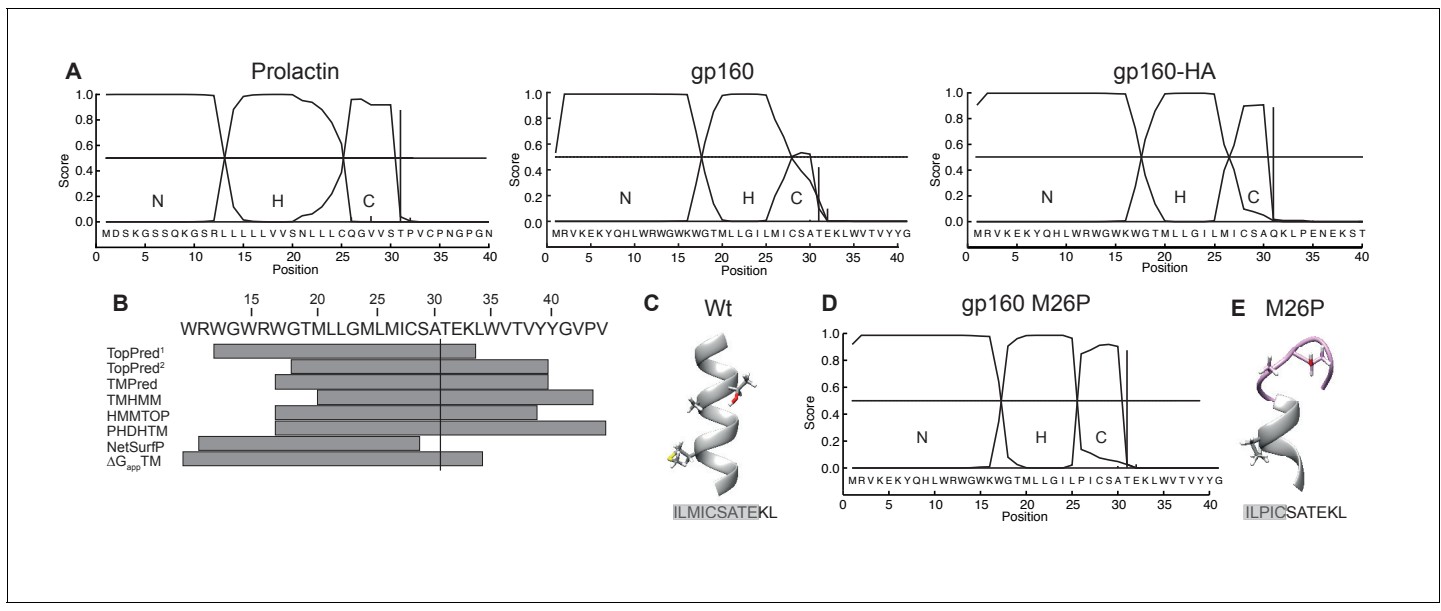

**Figure 6.** The C region of the gp160 signal peptide overlaps with the hydrophobic core region. (**A**) Signal-peptide prediction tool SignalP 3.0 (36) was used to assess signal peptides of prolactin, gp160, and gp160 signal peptide followed by HA (as used in *Figure 1B*). The characteristic N-terminal charged region (N), the hydrophobic membrane-spanning region (H), and the C-terminal region (C) containing the cleavage site were plotted. Vertical bars represent the first amino acid after the cleavage site. (**B**) TopPred[1] (Goldman, Engelman, and Steitz scale), TopPred[2] (Kyte and Doolittle scale), TMPred, TMHMM, HMMTOP, PHDHTM, and NetSurf Helix were used to predict the transmembrane domain of the gp160 signal peptide. The predicted transmembrane domains are represented by grey bars below the HXB2 sequence (residues 12–45). The cleavage site is marked by a vertical line. (**C**) Robetta (*Kim et al., 2004*) was used to predict the structure of the area around the signal-peptide cleavage site of wild-type gp160. Residues M26, A30 and T31 are shown as sticks. Alpha helices colored in grey. (**D**) SignalP prediction of gp160 M26P signal peptide. (**E**) as in (**C**) except M26P gp160 was used for the prediction. Structures shown are representative of the 5 predicted structures received from the Robetta server.
DOI: https://doi.org/10.7554/eLife.26067.009

The following figure supplement is available for figure 6:

**Figure supplement 1.** De-novo structure predictions of gp160 wild-type and M26P signal peptide.
DOI: https://doi.org/10.7554/eLife.26067.010

H- and C-region visibly improved and cleaved signal-peptide probability increased to a near perfect 0.936 (*Figure 6A*), consistent with our experimental findings (*Figure 1B*).

The SignalP predictions combined with our finding that T31 is at the lumen-membrane interface suggest that the membrane-spanning alpha helix of the gp160 signal peptide extends beyond position 31, allowing immersion into the membrane and occluding the cleavage site. Secondary structure is known to inhibit cleavage by proteases and could account for the initial resistance of the gp160 signal peptide to signal-peptidase activity (*Fluhrer et al., 2012*). Indeed, the majority of transmembrane-domain prediction algorithms suggested that residues 33 up to 44 of gp160 form an extended alpha helix (*Figure 6B*). De novo structure predictions using Robetta (*Kim et al., 2004*) also show the alpha helix overlapping with the cleavage site (*Figure 6C*). Together, these analyses suggest a model in which an extended α-helical structure around the cleavage site impairs accessibility to signal peptidase.

## Breaking secondary structure causes co-translational signal-peptide cleavage

If helical structure around the cleavage site is prohibitive, disruption of the helix would be predicted to lead to co-translational signal-peptide cleavage of gp160. Importantly, a point mutation in the signal peptide should not directly impact mature gp160, whose sequence is unaltered. To break secondary structure, we introduced a proline in the signal peptide close to the cleavage site. Prolines are common in signal peptides but are absent in 99.9% of known HIV-1 gp160 sequences (*Supplementary file 1*). SignalP analysis revealed that mutation M26P increased signal-peptide cleavage probability from 0.628 to 0.928 and clearly separated H- and C-regions (*Figure 6D*). Similarly, de novo structure predictions also showed a break in alpha-helical structure around the cleavage site (*Figure 6E*, *Figure 6—figure supplement 1*).

To study timing of signal-peptide removal and the potential impact on the folding and maturation of gp160, we analyzed the fate of M26P gp160 in our folding assay (*Figure 7A*). On a reducing gel (Cells R) wild-type gp160 with its signal peptide still attached (Ru) runs as a single band of ~100 kDa after synthesis (*Land et al., 2003*). From 15 min onward signal peptide-cleaved gp160 (Rc) appears just below Ru. Upon proper folding, trimerization, furin cleavage, and arrival at the plasma membrane, gp120 sheds from gp41 and is detected in the medium (*Moore et al., 1990*).

Immediately after synthesis M26P gp160 already ran primarily as a single band in the position of signal-peptide-cleaved gp160 (*Figure 7A*, Cells R). As predicted, cleavage occurred co-translationally. The overall signal of EndoH-sensitive cell-associated M26P gp160 and kinetics of gp120 shedding were comparable to wild-type type protein (*Figure 7A*). Even though M26P gp160 signal peptide was rapidly removed, the mutant took as long as wild-type gp160 to fold and exit the ER. We did not detect any difference in aggregation or degradation of wild-type versus M26P gp160.

A second, faster-migrating band of M26P (asterisk) was identified as cytosolic and attributed to suboptimal targeting to the ER due to weaker SRP binding: it did not change between reducing and non-reducing gels (*Figure 7A*, asterisk), or when DTT was removed from cells (*Figure 7B*, asterisk), indicating that it did not form disulfide bonds, it still contained the signal peptide, but did not contain the ~25 GlcNAc moieties that are left after deglycosylation with Endo H.

On non-reducing gel (Cells NR) the mobility of a protein is determined by its mass (as in reducing gels) as well as its compactness due to disulfide bonds (*Braakman et al., 1991*). Early gp160 folding intermediates (IT) run close to completely reduced gp160 and represent molecules with few or short-distance disulfide bonds (*Land et al., 2003*). Over time, disulfide bonds continue to form and isomerize, which leads to more compact folding intermediates. Between 15 and 30 min after synthesis, a distinct high-mobility band appears, representing gp160 that has folded properly and has achieved its native set of disulfide bonds (*Land et al., 2003*). Unexpectedly, co-translational signal-peptide cleavage of M26P gp160 significantly altered the oxidative folding pathway; folding intermediates acquired high compactness more quickly during synthesis (*Figure 7A*, Cells NR). Increased compactness is caused by a larger number of disulfide bonds or disulfides between more distant cysteines or both. Despite this increased rate of compaction, the appearance of the distinct native (NT) band and shedding of M26P gp120 from gp41 took as long as for wild-type gp160 (*Figure 7A*).

Because M26P loses its signal peptide co-translationally, it already begins with a mass difference of ~3 kDa compared to wild type. To ensure that this mass difference alone did not result in the more rapid folding phenotype on gel, we included 5 mM DTT in the pulse medium to postpone

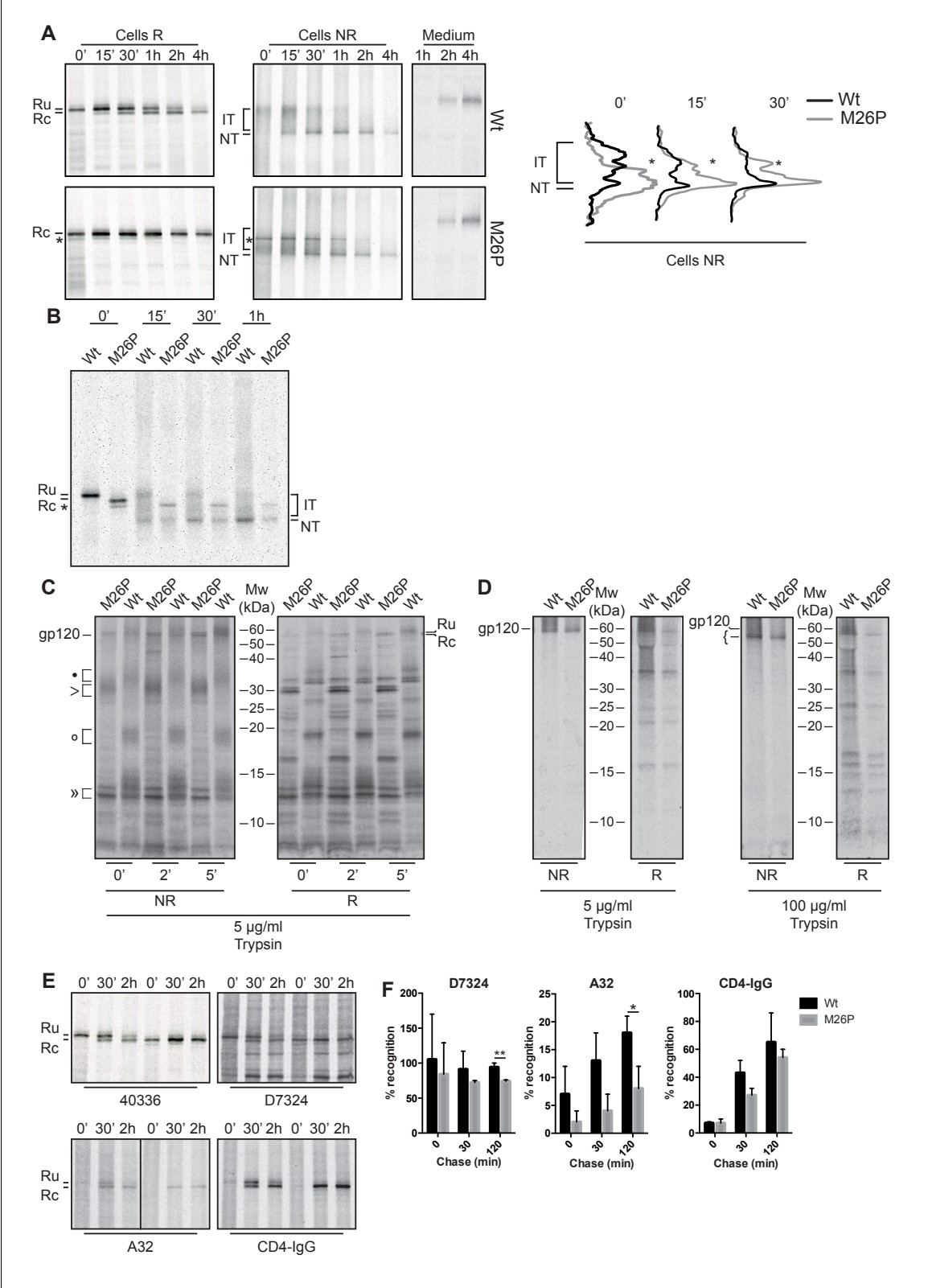

**Figure 7.** M26P leads to co-translational signal-peptide cleavage. Experiments were done as in *Figure 1*. (**A**) HeLa cells expressing wild-type and M26P gp160 were radiolabeled for 10 min and chased for the indicated times. Samples were deglycosylated with endoH and subjected to reducing (Cells R) and non-reducing (Cells NR) 7.5% SDS-PAGE. Medium samples were reduced and not deglycosylated. Lane profiles depicting the folding-intermediate (NR) smear of wild-type and M26P gp160 were determined from autoradiographs. (**B**) As in A except that wild-type and M26P gp120 were used and *Figure 7 continued on next page*

*Figure 7 continued*

samples were pulse labeled for 5 min in the presence of 5 mM DTT and chased in the absence of DTT. (**C + D**) HeLa cells expressing wild-type and M26P gp120 were pulse labeled as above and chased for either 0, 2, or 5 min (**C**) or 2 h (**D**). At the end of each time point, detergent cell lysates were proteolyzed with 5 or 100 μg/ml trypsin for exactly 15 min on ice. Proteolyzed samples were processed as in *Figure 1* and analyzed by 15% SDS-PAGE. (**E**) HeLa cells expressing wild-type or M26P gp160 were pulse labeled and chased as above. Detergent cell lysates were immunoprecipitated in parallel with either polyclonal antibody 40336 or antibodies A32, D7324, or CD4-IgG. After immunoprecipitation, samples were processed as in *Figure 1*. (**F**) Quantifications of experiments from E. Values were normalized compared to immunoprecipitation by 40336. Statistics were calculated using an unpaired t-test with Welch's correction. Exact p values can be found in *Figure 7—source data 1*. IT: folding intermediates; NT: native gp160; *: uncleaved unglycosylated M26P gp160 that had not targeted properly to the ER, likely due to its suboptimal signal sequence. Gels shown are representative of at least 3 independent experiments (biological replicates).
DOI: https://doi.org/10.7554/eLife.26067.011

The following source data is available for figure 7:

**Source data 1.** Autoradiographs from *Figure 7F* were quantified and each antibody normalized to the recognition of polyclonal antibody 40336 which recognizes all forms of gp160.
DOI: https://doi.org/10.7554/eLife.26067.012

disulfide-bond formation until the start of the chase (without DTT) (*Figure 7B*). Postponing disulfide-bond formation synchronized folding intermediates and diminished differences in co-translational folding that may exist between wild type and M26P. Again, M26P folding intermediates acquired high compactness more rapidly than wild-type gp160, with the majority of protein accumulating close to the native band after 15 min of chase. Taken together, these experiments support the model that secondary structure around the cleavage site inhibits co-translational signal-peptide cleavage.

Given the large differences in oxidative folding between wild type and M26P, we used limited proteolysis to examine conformational differences (*Hoelen et al., 2010*; *Kleizen et al., 2005*). In short, cells subjected to pulse-chase analysis were lysed in the absence of protease inhibitors and incubated on ice with 5 μg/ml trypsin for exactly 15 min. Protease inhibitors then were added and samples were immunoprecipitated and deglycosylated as before (*Figure 7A*) and analyzed by non-reducing (*Figure 7C* NR) and reducing (*Figure 7C* R) 15% SDS-PAGE.

At all 3 time points the digests from both wild-type and M26P gp120 contained a doublet of ~13 kDa (») whether reduced (R) or not (NR) (*Figure 7C*). Except for a small difference in definition these fragments were invariant, did not contain the N-terminus (as the signal peptide will have been cleaved from M26P but still is present in wild-type gp120) and did not contain detectable disulfide bonds. Differences did arise in the larger N-terminal fragments. With disulfide bonds intact (NR), M26P proteolysis produced a single diffuse band of ~30 kDa (>), whereas the wild-type gp120 digest contained three diffuse bands, at ~35 (•), ~20 kDa (○), and ~15 kDa. Upon reduction, all diffuse bands, of both wild type and M26P, dissociated into many well-defined bands (*Figure 7C* R) that must have been disulfide linked in the diffuse large fragments. The patterns of wild type and M26P were similar, albeit with a mobility shift explained by the presence of the signal peptide in the wild-type fragments, implying that these fragments were N-terminal. M26P was digested into a slightly larger number of fragments, indicative of subtle conformational differences and more conformational heterogeneity than wild-type gp120. The three disulfide-linked diffuse bands from wild-type gp120 indicated that the protease had cleaved between 3 clusters of relatively local disulfide bonds, most likely between the variable loops. In contrast, the single large band of M26P implies the presence of long-distance disulfide bonds between these 3 clusters. This is consistent with the increased compactness on non-reducing gel after the pulse (*Figure 7A*), as long-distance disulfide bonds would confer more compactness than short-distance ones.

## Co-translational signal-peptide cleavage causes localized misfolding in gp120

Because wild-type and M26P Envelope accumulated in the same 'native' position on SDS-PAGE gel, we extended the conformational assay from *Figure 7C* to both proteins after folding (*Figure 7D*). Strikingly, at the end of the 2-h chase both M26P and wild-type gp120 were highly protease resistant under non-reducing conditions (*Figure 7D*). Even when we increased the protease concentration to 100 μg/ml trypsin, we still immunoprecipitated a fragment of almost full-length size from

both proteins. As before, these large fragments dissociated into well-defined bands upon reduction (*Figure 7D*, R), indicating that in both natively-folded proteins the cleavage must have primarily occurred within the variable loops as this would allow disulfide bonds to hold the structure intact. As we found no significant difference in the fragments after reduction, we concluded that both wild-type and M26P contained a similar set of disulfide bonds and a similar conformation.

To further probe differences in the antigenic structures of wild-type and M26P gp160, we used conformation-specific antibodies. We pulse labeled samples as before and chased for 0, 30 min or 2 h before detergent lysis, immunoprecipitation, deglycosylation, and analysis by reducing SDS-PAGE (*Figure 7E*). Gp160 signals were quantified and normalized to recognition by polyclonal antibody 40336 (*Figure 7F*). We did not detect any significant differences at early chase times (0 and 30 min), likely due to the large heterogeneity in folding intermediates for gp160. Only when we examined the 2-h samples, which contained mostly 'native' gp160, did we detect significant differences between wild-type and M26P gp160. Antibody A32, which recognizes a 3D epitope in the inner domain of gp120 around disulfide 54–74, showed a significant decrease in recognition of M26P (*Figure 7F*). Misfolding was not limited to gp120, as antibody D7324, which recognizes an epitope near the disulfide bond in gp41, also showed a significant decrease in M26P recognition (*Figure 7F*). Given that the inner domain of gp120 interacts with gp41 in the trimer (*Garces et al., 2015*), it is possible that misfolding of the inner domain led to the misfolding in gp41. Immunoprecipitation with CD4-IgG did not show a difference between wild-type and M26P gp160 (*Figure 7F*). We concluded that co-translational signal-peptide cleavage resulted in localized misfolding in gp160. This misfolding however, was not sufficient to cause M26P to be retained in the ER or degraded.

## Co-translational signal-peptide cleavage of gp160 affects function

Because co-translational signal-peptide cleavage significantly altered the complex oxidative folding process of gp160, we anticipated the M26P mutation to affect gp160 function. We therefore assessed the impact of the M26P mutation on virus replication and fitness compared to the wild-type virus. SupT1 cells were transfected with 20 µg wild-type or mutant virus using electroporation, and viral titers were determined by CA-p24 ELISA at multiple time points (*Figure 8A*). M26P virus exhibited a slight delay in replication compared to wild-type virus, with virus levels being significantly lower at days 3 and 4, although the M26P mutant caught up with the wild-type virus at day 7. Because we expected that the maintenance of delayed signal-peptide cleavage across HIV-1 subtypes would confer a selective advantage, we tested the fitness of the M26P mutant virus compared to the wild-type virus, in a direct virus-competition assay. SupT1 cells were infected with a mix of wt: M26P LAI virus (50 pg in total) in two ratios, 1:1 and 1:10, both in duplicate cultures (*Figure 8B*). On days 1, 2, 3, 4 and 7 viral titers were monitored by CA-p24 ELISA (*Figure 8—figure supplement 1*); this was not continued after virus was passaged to new cells. After 4 and 35 days of culturing, the virus was harvested, sequenced and the electropherograms were quantified. When mixed at a 1:1 ratio the wild-type virus had outcompeted M26P virus by day 4 and the M26P mutant was only present as a minority variant at day 35. Moreover, when added in a ten-fold excess, the M26P mutant dominated the population at day 4, but again only formed a minority variant by day 35. These data indicate that the M26P mutant has a selective disadvantage to the wild-type virus.

We next assessed whether the M26P virus was less infectious than the wild-type virus on TZM-bl target reporter cells. Although not statistically significant, a trend was observed that the M26P mutant was less infectious, consistent with the competition data. We also noticed a greater variability in the M26P samples (*Figure 8C*). Possibly, the increased rate of attaining the native conformation led to larger heterogeneity in Env conformations, consistent with the increased fidelity of folding when folding rates are delayed by chaperones or lower temperature (*Daniels et al., 2003*; *Hebert et al., 1996*; *Denning et al., 1992*; *Sekhar et al., 2012*; *Sherman and Qian, 2013*).

To examine the effect of co-translationally cleaved gp160 on viral infectivity in the absence of other HIV factors, and to assess the effect in the context of a different isolate, we moved to a pseudovirus system using JR-FL gp160. TZM-bl cells were infected with 1000 pg of JR-FL wild-type or M26P pseudo-virus and infectivity was measured using a luciferase reporter assay. While wild-type JR-FL pseudo-virus readily infected target cells, M26P pseudo-virus was almost non-infectious (*Figure 8D*). We also compared the infectivity of LAI pseudo-virus containing the Ig κ and cystatin signal peptides (*Figure 8E*) and found them to be significantly less infectious than wild-type LAI pseudo-virus. Taken together, our results suggest that delayed cleavage of the signal peptide

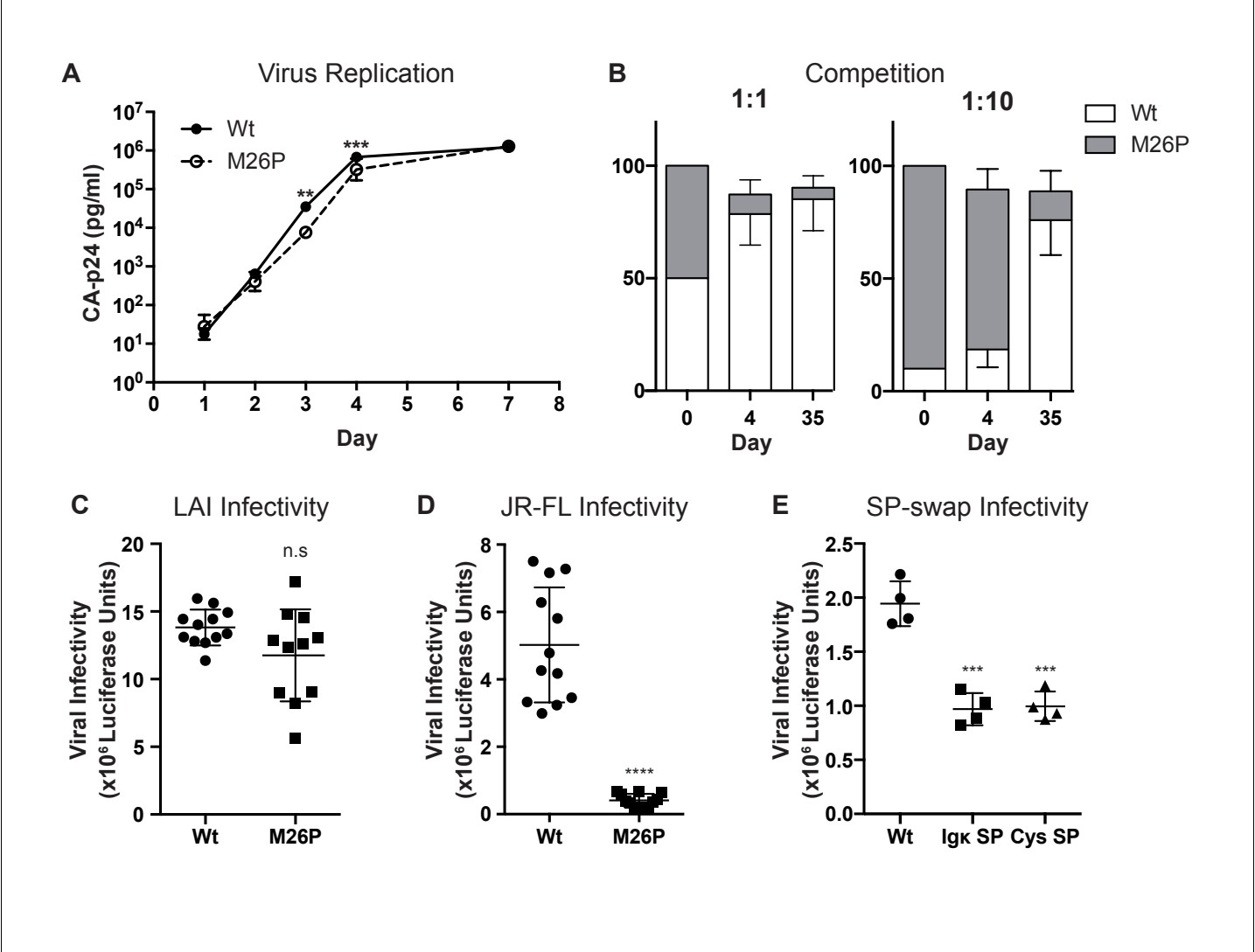

**Figure 8.** Co-translational signal-peptide cleavage causes functional gp160 defects. (**A**) SupT1 cells (5x10[6]) were infected with 20 µg of wild-type or M26P-mutant pLAI DNA constructs using electroporation. Replication was monitored by CA-p24 ELISA. The replication curves shown represent the averaged values of 6 independent experiments. p=0.0058 (**) and 0.003 (***). (**B**) SupT1 cells (*Krogh et al., 2001*) were infected with a total of 50 pg CA-p24 equivalent wild-type or M26P-mutant LAI virus (produced by transfected HEK293T cells) in ratios 1:1 and 1:10, each in duplicate (biological replicate). After 4 and 35 days, the isolated virus from each culture was sequenced and the electropherograms were quantified. (**C,D,E**) TZM-bl reporter cells were infected with: 500 pg of wild-type or M26P LAI virus (produced by HEK293T cells) (n.s, p=0.0833) (**C**), 1,000 pg of wild-type or M26P-mutant JR-FL pseudo-virus (produced by HEK293T cells) (p<0.0001). One M26P data point was excluded as it lay well below the background. (**D**), 1,000 pg of wild-type, Ig κ (p=0.0004) or cystatin-SP (p=0.0005) mutant LAI pseudo-virus (produced by C33A cells) (**E**) and infectivity was measured by Luciferase activity. The data in panels **A, C** and **D** are derived from 3 independent experiments, using 3 independently produced (pseudo-)virus stocks (biological replicates), each performed in quadruplicate (technical replicates). The data in panel **B** are derived from 2 independent experiments, using 2 independently produced virus stocks (biological replicates) and sequenced with two independent primers (technical replicates). The data in panel **E** are derived from 1 pseudovirus stock (biological replicate), performed in quadruplicate (technical replicates). All significance values were calculated with an unpaired, two-tailed t test with Welch's correction.

DOI: https://doi.org/10.7554/eLife.26067.013

The following figure supplement is available for figure 8:

**Figure supplement 1.** Viral replication in competition assays.

DOI: https://doi.org/10.7554/eLife.26067.014

contributes to maximizing production of functional gp160 and that co-translational signal-peptide cleavage results in fast folding of lower-quality gp160.

## Discussion

We found that the dramatically delayed cleavage of the signal peptide of HIV-1 gp160 functions as folding regulator to ensure correct formation of disulfide bonds and functional structure. This delayed cleavage is evolutionarily conserved across HIV-1 subtypes and is important for viral function. Secondary structure in the form of an alpha helix extends through the cleavage site preventing early access and cleavage by signal peptidase. Uncleaved gp120 hence transiently resides in the ER as a signal-anchored type-II transmembrane protein with the cleavage site shielded by membrane phospholipids. The integral-membrane intermediate is retained in the ER, and we here suggest that the signal peptide acts a quality-control mechanism to ensure that gp160 exits the ER in an optimally folded form. Forced co-translational cleavage by insertion of a helix-breaking proline in the signal peptide led to changes in gp120 folding and reduced Env function.

Residues flanking the cleavage site influence removal of a signal peptide as the ER signal peptidase does not limit its activity to a strict consensus sequence (*Choo and Ranganathan, 2008*; *Li et al., 1988*; *Andrews et al., 1988*). While this enzyme may be a specific protease in terms of its location and topology, the signal peptidase nonetheless does show hallmarks of a typical protease: structural descriptors for cleavage are exposure, flexibility, and local interactions (*Hubbard et al., 1991*; *Kazanov et al., 2011*; *Novotný and Bruccoleri, 1987*; *Overall, 2002*; *Timmer et al., 2009*). Proteases prefer loops and disfavor α-helices and β-sheets. The helix-breaking M26P mutation supports our conclusion that lack of co-translational Env signal-peptide cleavage is primarily the result of secondary structure around the cleavage site. NMR studies of *E.coli* signal peptidase in complex with alkaline phosphatase signal peptide revealed that the cleavage region adopted a poorly structured 'U-turn' shape (*De Bona et al., 2012*). The loop originated from proline in position −5 to the cleavage site confirming the role of proline in separating hydrophobic and C-terminal region of signal peptides by inducing formation of unstructured turns or loops predicted in the literature earlier (*von Heijne, 1983*; *Jain et al., 1994*). Enrichment of proline near signal-peptide cleavage sites is not limited to *E. coli*: a database of all verified signal peptides from Archaea to mammals shows ~20% of all cleavable signal peptides containing at least one proline in the −1 to −5 position relative to the cleavage site (*Choo et al., 2005*). Gp160 sequences are virtually devoid of proline in that position (*Table 1*). Ninety-five percent of all analyzed sequences contain M26 or I26 with only 5 sequences out of >4100 containing a proline in the −1 to −5 position (*Table 1*, *Supplementary file 1*). In addition to helical structure, a dearth of water at the lipid-immersed cleavage site hinders signal-peptidase activity, as all hydrolysis reactions by definition require water.

The SignalP algorithm used in this study is trained to distinguish between cleavable signal peptides and uncleavable signal anchors (*Bendtsen et al., 2004*; *Nielsen et al., 1997*). We show here that a signal peptide with mixed probabilities may well function as a transient signal anchor. A sequence with similar signal-peptide/anchor probabilities hence may suggest a similar 'bipolar' anchor/peptide behavior, provided that targeting is efficient and does not affect the probability computation. The distinction between a signal peptide that is engaged by the translocon and a signal anchor immersed in the lipid bilayer is subtle because the lateral gate of the translocon allows partial immersion in the bilayer within the translocon (*Li et al., 2016*; *Gogala et al., 2014*; *Voorhees et al., 2014*). This confirms earlier biochemical data demonstrating lipid interactions of the nascent chain in the translocon (*Martoglio et al., 1995*; *Martoglio and Dobberstein, 1995*; *Do et al., 1996*; *Higy et al., 2004*).

During folding, the transient signal anchor of gp120 tethers it to the membrane, restricting conformational freedom of the N-terminus. Tethering of the protease domain of the Semliki Forest virus capsid protein to the membrane via a non-cleavable signal anchor resulted in more efficient folding (*Kowarik et al., 2002*), and we propose that tethering gp120 might be similarly beneficial for gp120 folding. In-silico folding simulations of knotted proteins demonstrated that efficiency of knot-domain formation was enhanced by tethering of the N-terminus (*Soler and Faísca, 2012*). Indeed, many proteins have their N-termini restrained by ribosome-associated chaperones such as *E.* coli trigger factor (*Kaiser et al., 2006*; *Kaiser et al., 2011*), which prevent early, non-productive folding interactions. In the context of gp160 this can be illustrated by our finding that the M26P mutant

**Table 1.** M26 and I26 have highest conservation in gp160 sequences An alignment of 4236 gp160 sequences (**Supplementary file 1**) was used to compare absolute occurrence of each amino acid in position 26 of the signal peptide (SP) and the respective cleavable SP probability predicted by SignalP 3.0.

| Position 26 | SP probability | Absolute occurrence |
|---|---|---|
| Y | 0.458 | 0 |
| I | 0.51 | 1375 |
| F | 0.591 | 7 |
| V | 0.604 | 16 |
| **M** | **0.628** | **2638** |
| L | 0.697 | 143 |
| C | 0.699 | 1 |
| W | 0.751 | 1 |
| T | 0.777 | 39 |
| G | 0.792 | 0 |
| A | 0.805 | 2 |
| S | 0.848 | 5 |
| K | 0.863 | 0 |
| Q | 0.897 | 0 |
| E | 0.901 | 4 |
| D | 0.906 | 0 |
| R | 0.909 | 3 |
| N | 0.923 | 0 |
| H | 0.928 | 1 |
| P | 0.928 | 1 |

DOI: https://doi.org/10.7554/eLife.26067.015

immediately collapses into a more compact, disulfide-bonded state leading to a substantially more heterogeneously folded gp160 population. Post-translational cleavage of gp160 restricts conformational freedom, thereby preventing the initial collapse, and increases folding fidelity while ultimately allowing release from its tether, which is essential for its role in viral infectivity.

Previously, we have shown that signal-peptide cleavage requires folding and N-glycosylation of gp120 (**Land et al., 2003**), and that five out of ten conserved disulfide bonds are required for cleavage (**van Anken et al., 2008**). Crystal structures of gp120 show that it adopts a hairpin conformation, with its N- and C-termini in close proximity (**Garces et al., 2015**). Given that the far N-terminus is a β-strand in the native protein whereas our studies have suggested this region to be α-helical early on (which prevents co-translational cleavage), a late folding event (possibly integration of the N- and C-termini) likely triggers the conformational change in the N-terminus, allowing cleavage to occur. This is underscored by the poor cleavage of the various reporter constructs that lacked the C-terminal residues of gp120 necessary to initiate this conformational change.

The M26P mutation had a dramatic impact on infectivity of JR-FL pseudovirus. However, considering the conservation of post-translational cleavage, one might have expected a more pronounced effect of the M26P mutation on LAI virus infectivity. Although wild-type LAI readily outcompeted the co-translationally cleaved M26P Env-containing virus, the difference in infectivity of target cells was subtle. The defects caused by co-translational cleavage may be much larger in vivo.

As the association between subunits gp120 and gp41 involves gp120's N-terminus, it is appropriate that this conserved interface suffered from early cleavage; we found that co-translational signal-peptide cleavage caused localized misfolding in gp41 and in the inner domain of gp120. Because the stability of the gp120/gp41 association differs substantially between HIV-1 strains this may explain the more dramatic impact of the M26P mutation in the context of the JR-FL isolate versus LAI. Strain-specific differences in stability of subunit association may be the reason why the effect of

a heterologous signal peptide on gp160 function has been controversial thus far. The gp160 signal peptide was often replaced without detected negative effects (*Li et al., 1996*; *Lasky et al., 1987*; *Li et al., 1994*) whereas in other cases replacement led to decreased infectivity (*Pfeiffer et al., 2006*). Heterologous signal peptides may compensate for functional defects by increased expression of gp160 and the resulting increase in incorporation into new virions. Although as few as 9–14 native trimers on the virion surface suffice for infection (*Zhu et al., 2006*; *Klasse, 2007*), suboptimal Env trimers may require increased density for function.

Despite great sequence variability, signal peptides of different secretory proteins have been considered interchangeable. While this may be true for their primary role, ER targeting, it becomes increasingly obvious that signal peptides fulfill diverse other functions starting from the moment they emerge from the ribosome. Hydrophobicity of prokaryotic signal peptides (*Valent et al., 1997*; *Lee and Bernstein, 2001*) or in yeast proteins (*Ng et al., 1996*) determines whether protein translocation is SRP-dependent or independent and generally the targeting efficiency of a signal peptide regulates expression levels. Gp160 for instance shows increased expression over the course of infection when the N-terminal charges in the signal peptide decrease (*Li et al., 1994*; *da Silva et al., 2011*; *Asmal et al., 2011*; *Gnanakaran et al., 2011*). Low efficiency in targeting may result in dual localization of the mature protein, which in the case of calreticulin gives rise to functional ER-lumenal and cytosolic pools (*Shaffer et al., 2005*). Actively tuning translocation efficiency of distinct sets of proteins allows the cell to react to ER stress (*Kang et al., 2006*). To lower the burden on the ER, newly synthesized proteins that fail to translocate are degraded in the cytosol and inefficient SRP binding triggers Argonaute2-dependent mRNA degradation (*Kang et al., 2006*; *Karamyshev et al., 2014*).

In addition to targeting, signal peptides may influence topology of the mature protein, as was shown for the human prion protein, PrP (*Ott and Lingappa, 2004*; *Hegde et al., 1998*), or they may regulate folding [e.g. gp160, US11, and EspP (*Rehm et al., 2001*; *Szabady et al., 2005*)]. In this sense, they are specialized, membrane-bound versions of soluble pro-peptides that function as intramolecular chaperones before their removal. They function in places that cellular chaperones cannot reach. These soluble pro-peptides may contain a cysteine that supports intramolecular disulfide-bond isomerization as seen with bovine pancreatic trypsin inhibitor (*Weissman and Kim, 1992a*, *1992b*) or a specific sequence interacting with the folding protein (insulin). Signal peptides immersed in the membrane cannot fulfill these roles very robustly as most of the signal peptide is not available for interaction, but in essence the functions of soluble pro-peptides and intramembrane signal peptides may be similar.

The more we learn about different functions of signal peptides the more understandable their variability. Every function requires a regulatory mechanism, and soluble pro-peptides may set the example. We do conclude that rather than their blunt exchange, signal peptides do deserve attention, as they are not as inert as often anticipated.

## Materials and methods

### Plasmids, antibodies, reagents, and viruses

The full-length molecular clone of HIV-1$_{LAI}$ (pLAI) was the source of wild-type and mutant viruses (*Peden et al., 1991*). The QuikChange Site-Directed Mutagenesis kit (Stratagene) was used to introduce mutations into *env* in plasmid pRS1 as described before; the entire *env* gene was verified by DNA sequencing (*Sanders et al., 2004*). Mutant *env* genes from pRS1 were cloned back into pLAI as SalI-BamHI fragments.

For transient transfection of gp120/160 we modified pcDNA3 by introducing an Intron A sequence upstream of the start ATG (*Chapman et al., 1991*), and named the plasmid pMQ. Influenza A virus hemagglutinin (HA, from avian H3N2 HA/Aichi/68) was also subcloned into pMQ. All mutants were generated from wild-type gp120 or gp160 with the corresponding primer pairs using QuikChange. Non-native signal peptides were purchased as gene fragments (IDT) and inserted into pMQ gp120 digested with XbaI and KpnI using Gibson Assembly (*Gibson et al., 2009*) or pMQ X31-HA digested with XbaI and HindIII. For immunoprecipitations we used the previously described polyclonal rabbit anti-gp160 antibody 40336 (*Land et al., 2003*) and 'P', polyclonal rabbit serum raised by the Braakman lab against purified HA and characterized to recognize folded, misfolded,

and denatured HA. Although we studied gp160 of the LAI isolate, we followed the canonical HXB2 residue numbering (GenBank: K03455.1), which relates to the LAI numbering as follows: because of an insertion of five residues in the V1 loop of LAI gp160, all cysteine residues beyond this loop have a number that is 5 residues lower in HXB2 than in LAI: until Cys131, numbering is identical, but Cys162 in LAI becomes 157 in HXB2, etc.

GFP constructs ER-RFP, ER-GFP, ER-GFP-KDEL and gp120 sfGFP have been previously described (*Costantini et al., 2015*; *Snapp et al., 2006*; *Aronson et al., 2011*). This construct was used as the backbone for all of the different SP-GFP constructs. The HIV-1 gp160 SP constructs included the SP and one to thirty post-SP amino acids (+1 to +30). Mutants were created either in PCR 3' reverse oligos or using the QuikChange kit from Stratagene, as recommended by the manufacturer.

For the membrane integration assay an ApaI site was introduced after the N-terminal 89 residues of HXB2 gp160, expressed from pcDNA3 (*van Anken et al., 2008*). To create a Lep P2-domain fusion protein an ApaI site at position 85–86 and an XhoI site at the 3′-end were introduced into *lepB*. The P2 domain of Lep containing a natural glycosylation site at position 215–217 (NET) was introduced as ApaI-XhoI fragment. Glycosylation acceptor sites (NST) were inserted at appropriate positions in gp160 (N39, N40, N41 and N47) by site-specific mutagenesis (QuikChange Site-Directed Mutagenesis Kit, Stratagene). All inserted fragments and mutants were confirmed by sequencing at Eurofins MWG Operon (Ebersberg, Germany).

## Cells and transfections

The SupT1 cell line (ATCC CRL-1942, RRID:CVCL_1714) was cultured in Advanced RPMI 1640 medium (Gibco), supplemented with 1% fetal calf serum (v/v, FCS), 2 mM L-glutamine (Gibco), 15 units/ml penicillin and 15 µg/ml streptomycin. The TZM-bl reporter cell line (Cat# 8129–442, RRID: CVCL_B478), obtained from NIH AIDS Research and Reference Reagent Program, Division of AIDS, NIAID, NIH (John C. Kappes, Xiaoyun Wu, and Tranzyme, Inc., (Durham, NC)), the HEK293T cell line (ATCC CRL-3216, RRID:CVCL_0063), and the C33A cell line (ATCC HTB-31, RRID:CVCL_1094) were cultured in Dulbecco's modified Eagle medium (Gibco) containing 10% FCS, 100 units/ml penicillin and 100 µg/ml streptomycin. HeLa cells (ATCC CRL-7924, RRID:CVCL_0058) were maintained in MEM containing 10% FCS, nonessential amino acids, glutamax and penicillin/streptomycin (100 U/ ml). Twenty-four hours before pulse labeling HeLa cells were transfected with pMQ gp120/gp160 or HA constructs using polyethylenimine (Polyscience) as described before (*Hoelen et al., 2010*). COS-7 cells (ATCC CRL-1651, RRID:CVCL_0224) were grown in RPMI lacking phenol red plus glutamine, 10% heat-inactivated FCS, and penicillin/streptomycin. All cell lines were maintained at 37°C with 5% $CO_2$ and routinely tested negative for mycoplasma contamination. All cell lines were assumedly authenticated by their respective sources and were not further authenticated for this study.

## Virus production

Virus stocks were produced by transiently transfecting HEK293T cells with wild-type or mutant pLAI constructs using the Lipofectamine 2000 Transfection Reagent (Invitrogen) according to the manufacturer's protocol. Alternatively, virus stocks were produced by transfecting C33A cells by calcium-phosphate precipitation (*van Anken et al., 2008*).The virus-containing culture supernatants were harvested 2 days post-transfection, stored at −80°C, and the virus concentrations were quantitated by CA-p24 ELISA as described previously (*Moore and Jarrett, 1988*). These values were used to normalize the amount of virus used in subsequent infection experiments.

## Virus replication

A total of $5 \times 10^6$ SupT1 cells were transfected with 20 µg wild-type or M26P mutant pLAI DNA constructs using electroporation. Virus spread was monitored for 8 days, by visual inspection for the appearance of syncytia and by CA-p24 ELISA as indicators of virus replication.

## Competition assay

A total of $10^5$ SupT1 cells were infected with a total of 50 pg LAI virus (produced on HEK293T cells). The wild-type:M26P sample in a 1:1 ratio contained 25 pg virus each. For the 1:10 ratio wild-type: M26P sample, 5 pg wild-type virus was combined with 45 pg M26P mutant virus. Virus spread was monitored by visual inspection for the appearance of syncytia and by CA-p24 ELISA as indicators of

virus replication. Decreasing amounts of supernatant were passaged when the cells were (almost) wasted due to infection by the replicating virus. Viruses were cultured for 35 days and passaged cell-free onto uninfected SupT1 cells when virus replication was apparent. Virus replication was quantitated for the first 7 days by CA-p24 ELISA. On days 4 and 35 cellular DNA was extracted from infected cells using the QIAamp DNA Mini kit (Qiagen) according to the manufacturer's instructions and the complete env genes from proviral DNA sequences were PCR-amplified using the Expand High Fidelity PCR System (Roche) as described before (*Eggink et al., 2008*). DNA sequences then were sequenced using forward and reverse primers and the peak heights extracted from the electropherograms were determined using the ab1PeakReporter utility (Life Technologies).

## Single cycle infection

The TZM-bl reporter cell line stably expresses high levels of CD4 and HIV-1 coreceptors CCR5 and CXCR4 and contains the luciferase and β-galactosidase genes under the control of the HIV-1 long-terminal-repeat (LTR) promoter (*Wei et al., 2002*). Single-cycle infectivity assays were performed as described before (*Bontjer et al., 2009*; *Bontjer et al., 2010*). In brief, one day prior to infection, 17 × 10^6 TZM-bl cells per well were plated on a 96-well plate in DMEM containing 10% FCS, 100 units/ml penicillin and 100 µg/ml streptomycin and incubated at 37°C with 5% $CO_2$. A fixed amount of virus LAI virus (500 pg of CA-p24) or a fixed amount of JR-FL or LAI pseudo-virus (1,000 pg of CA-p24) was added to the cells that were at 70–80% confluency in the presence of 400 nM saquinavir (Roche) to block secondary rounds of infection and 40 µg/ml DEAE in a total volume of 200 µl. Two days post infection, the medium was removed, cells were washed with phosphate-buffered saline (50 mM sodium phosphate buffer, pH 7.0, 150 mM NaCl) and lysed in Reporter Lysis buffer (Promega). Luciferase activity was measured using a Luciferase Assay kit (Promega) and a Glomax luminometer (Turner BioSystems) per the manufacturer's instructions. Uninfected cells were used to correct for background luciferase activity. All infections were performed in quadruplicate.

## Folding assay

HeLa cells transfected with wild type or mutant gp160/gp120 or HA constructs were subjected to pulse-chase analysis as described before (*Land et al., 2003*). In short, cells were starved for cysteine and methionine for 15–30 min and pulse labeled for 5 min with 55 µCi/ 35 mm dish of Express $^{35}$S protein labeling mix (Perkin Elmer). Where indicated (+DTT) cells were incubated with 5 mM DTT for 5 min before and during the pulse. The pulse was stopped and chase started by the first of 2 washes with chase medium containing an excess of unlabeled cysteine and methionine. At the end of each chase, medium was collected and cells were cooled on ice and further disulfide bond formation and isomerization was blocked with 20 mM iodoacetamide. Cells were lysed and detergent lysates and medium samples were subjected to immunoprecipitation with polyclonal antibody 40336 against gp160 or polyclonal rabbit serum P against HA.

## Deglycosylation, SDS-PAGE, and autoradiography

To identify gp160 folding intermediates, glycans were removed from lysate-derived gp120 or gp160 with Endoglycosidase H (Roche) treatment of the immunoprecipitates as described before (*Land et al., 2003*). Samples were subjected to non-reducing and reducing (25 mM DTT) 7.5% SDS-PAGE. Gels were dried and exposed to super-resolution phosphor screens (FujiFilm) or Kodak MR films (Kodak). Phosphor screens were scanned with a Typhoon FLA-7000 scanner (GE Healthcare Life Sciences) and quantifications performed in ImageQuantTL (RRID:SCR_014246) and graphs prepared with Graphpad (RRID:SCR_000306).

## Light microscopy

Images were collected on a Zeiss Axiovert 200 with a 63x oil 1.4NA Planapo objective and a QImaging Retiga 2000R CCD. Imaged cells were grown in 8-well Lab-Tek chambers (Nunc). Images were collected with Qimaging software and processed using Adobe Photoshop CS2. All figures were prepared using Adobe Photoshop CS2 and Adobe Illustrator CS.

## Carbonate extraction

A 10 cm dish of transiently transfected Cos7 cells was scraped into 1 ml hypotonic buffer solution (10 mM Tris-HCl, pH 7.5 plus Boehringer-protease inhibitor mini-tablets) and dounce homogenized. Homogenates were spun first at 8000 *xg* to clear nuclei and debris, then at 100,000 *xg* for 1 hr and the resulting pellet then was extracted in sodium carbonate buffer (0.2 M sodium carbonate pH 11.5) for 30 min on ice followed by a second centrifugation at 100,000 *xg*. The extract and membrane pellet then were mixed with SDS-PAGE sample buffer to create equal volumes of extract supernatant and membrane solutions and analyzed by 12% tris-tricine SDS-PAGE and Western Blot.

## Fluorescence recovery after photobleaching (FRAP)

FRAP and fluorescence loss in photobleaching were performed by photobleaching a small ROI and monitoring fluorescence recovery or loss over time, as described previously (*Siggia et al., 2000*; *Snapp et al., 2003*). Fluorescence intensity plots and D measurements were calculated as described previously (*Siggia et al., 2000*; *Snapp et al., 2003*). To create the fluorescence recovery curves, the fluorescence intensities were transformed into a 0–100% scale and were plotted using Kaleidagraph 3.5 (RRID:SCR_014980). The p values were calculated using a Student's two-tailed t-test in Excel (Microsoft) or Graphpad (RRID:SCR_000306). Composite figures were prepared using Photoshop (RRID:SCR_014199) and Illustrator (RRID:SCR_014198) software (Adobe).

## In vitro translation

Constructs cloned into pcDNA3 were transcribed and translated in the TNT T7 Quick Coupled transcription-translation system (Promega). Using 10 µl of reticulocyte lysate and 150–200 ng of DNA template, 1 µl of L-[$^{35}$S]-Met (5 µCi) and 1 µl of EDTA-stripped, nuclease-treated dog pancreas rough microsomes (RM) were added at the start of the reaction, and samples were incubated for 90 min at 30°C. Translation products were analyzed by SDS-PAGE, and gels were visualized and quantified on a Fuji FLA-3000 PhosphorImager (Fuji film) with Image Reader 8.1J/Image Gauge software. The MultiGauge (Fujifilm) software was used to generate a profile of each gel lane and to calculate the peak areas of the glycosylated protein band.

## SignalP

SignalP 3.0 Server was used to predict signal-peptide probabilities of HXB2 gp160 wild type and mutants, X31 HA, and prolactin using a hidden Markov model (*Bendtsen et al., 2004*; *Nielsen et al., 1997*).

## TM predictions

The HXB2 sequence of gp160's 70 N-terminal residues was used to identify predicted TM spans with algorithms HMMTOP (*Tusnády and Simon, 2001*; *Tusnády and Simon, 1998*), TMPred (*Hofmann and Stoffel, 1993*), TMHMM (*Krogh et al., 2001*; *Sonnhammer et al., 1998*), TopPred (*Claros and von Heijne, 1994*; *von Heijne, 1992*), PHDhtm (*Rost and Sander, 1993*), $\Delta G_{app}$ TM full protein scan with TM lengths of 12–24 residues (*Hessa et al., 2007*), and NetSurfP (*Petersen et al., 2009*).

## Acknowledgements

We thank Dr Manu Hegde (Cambridge, UK), Prof. Maarten Egmond (Utrecht University) and members of the Braakman lab for valuable discussions and suggestions. This work was supported by grants from the Netherlands Organization for Scientific Research (NWO)-Chemical Sciences (IBr, AL, NM), the Netherlands AIDS Fund (IBr, AL), the European Union 7th framework program, ITN 'Virus Entry' (IBr, MQ, NM), and by the Center for AIDS Research at the Albert Einstein College of Medicine and Montefiore Medical Center funded by the National Institutes of Health (NIH AI-51519) (ELS, ZC), the Swedish Cancer Foundation (CK, IMN, GvH), and the Knut and Alice Wallenberg Foundation (GvH). RWS is a recipient of a Vidi grant from NWO and a Starting Investigator Grant from the European Research Council (ERC-StG-2011–280829-SHEV).

# Additional information

## Competing interests

Erik Lee Snapp: Has filed a patent application with and licensed technology to Lucigen Corp (U.S. Patent Application 15/152/908). The technology is not related to this manuscript. Rogier W Sanders: Is listed as an inventor on patents involving recombinant, soluble native-like Env trimers (EP2975053A1, EP2765138A3, WO/2017/055522A1, WO/2011/108937, WO/2010/041942, WO/2008/103428A2, WO/2003/022869A2). The technology is not related to this manuscript. The other authors declare that no competing interests exist.

## Funding

| Funder | Grant reference number | Author |
| --- | --- | --- |
| Nederlandse Organisatie voor Wetenschappelijk Onderzoek | | Nicholas McCaul<br>Matthias Quandte<br>Aafke Land<br>Ineke Braakman |
| Netherlands AIDS Fund | | Aafke Land |
| Seventh Framework Programme | ITN 'Virus Entry' | Nicholas McCaul<br>Matthias Quandte<br>Ineke Braakman |
| National Institutes of Health | NIH AI-51519 | Erik Lee Snapp |
| Swedish Cancer Foundation | | IngMarie Nilsson<br>Gunnar von Heijne |
| Knut och Alice Wallenbergs Stiftelse | | Gunnar von Heijne |
| European Research Council | ERC-StG-2011-280829-SHEV | Rogier W Sanders |

The funders had no role in study design, data collection and interpretation, or the decision to submit the work for publication.

## Author contributions

Erik Lee Snapp, Conceptualization, Formal analysis, Supervision, Funding acquisition, Investigation, Visualization, Methodology, Writing—review and editing; Nicholas McCaul, Conceptualization, Formal analysis, Supervision, Funding acquisition, Investigation, Visualization, Methodology, Writing—original draft, Writing—review and editing; Matthias Quandte, Conceptualization, Investigation, Visualization, Writing—original draft, Writing—review and editing; Zuzana Cabartova, Investigation, Visualization; Ilja Bontjer, Formal analysis, Investigation, Visualization, Writing—review and editing; Carolina Källgren, Investigation, Visualization, Writing—review and editing; IngMarie Nilsson, Aafke Land, Investigation, Visualization, Methodology; Gunnar von Heijne, Supervision, Funding acquisition, Methodology, Writing—review and editing; Rogier W Sanders, Supervision, Funding acquisition, Investigation, Methodology, Writing—review and editing; Ineke Braakman, Conceptualization, Supervision, Funding acquisition, Investigation, Writing—original draft, Writing—review and editing

## Author ORCIDs

Gunnar von Heijne https://orcid.org/0000-0002-4490-8569
Ineke Braakman http://orcid.org/0000-0003-1592-4364

## Decision letter and Author response

Decision letter https://doi.org/10.7554/eLife.26067.017
Author response https://doi.org/10.7554/eLife.26067.018

## Additional files

**Supplementary files**

• Supplementary file 1. Amino acid and N-linked glycan conservation across gp160 sequences. The 2014 filtered alignment of >4100 gp160 sequences was downloaded from the Los Alamos National Laboratory HIV-1 molecular database (www.hiv.lanl.gov) and placed into Microsoft Excel to calculate amino acid conservation at each position.

DOI: https://doi.org/10.7554/eLife.26067.016

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
