## [Decision Letter]

Thank you for submitting your article "Structure and topology around the cleavage site regulate post-translational cleavage of the HIV-1 gp160 signal peptide" for consideration by *eLife*. Your article has been favorably evaluated by Ivan Dikic (Senior Editor) and three reviewers, one of whom is a member of our Board of Reviewing Editors. The reviewers have opted to remain anonymous.

The reviewers have discussed the reviews with one another and the Reviewing Editor has drafted this decision to help you prepare a revised submission.

Summary:

The majority of the proteins that are targeted to the mammalian secretory pathway contain cleavable N-terminal signal sequences, yet little is known about the timing of cleavage and especially how signal sequence cleavage affects overall maturation of cargo proteins. This manuscript from Snapp et al., comprehensively looks at the timing of signal sequence cleavage for the signal sequence of gp160, and how its known delayed cleavage affects folding/oxidation and the fitness of viral progeny. A combination of bioinformatic tools and protein structure prediction methods suggest that the formation of an α-helical segment, that includes the signal sequence and the N-terminus of the mature protein, yields a signal-anchor sequence that tethers gp120 to the membrane during protein folding. The addition of a single helix breaking Pro residue (M26P) supports domain separation as determined using an in silico analysis and permits co-translational cleavage. Importantly, early cleavage results in rapid oxidation and the generation of weakly infectious virions. The authors propose a thoughtful and provocative model whereby the uncleaved signal sequence acts as an intramolecular chaperone to promote proper folding of gp160. Given the poor understanding and the importance of signal sequence cleavage to the fidelity of the maturation process and its widespread implications, this manuscript would be of significant relevance to a broad audience. Overall the manuscript is well written and the experiments are properly designed and interpreted. The reviewers concluded that a revised version of this manuscript would be suitable for *eLife*.

Essential revisions:

1) The authors would like to conclude that the altered folding kinetics of the M26P mutant yields a more heterogeneous population of gp160 proteins that would include misfolded variants, hence the less-fit phenotype. Unfortunately, the authors lack direct evidence that misfolded variants are produced. While the M26P mutant appears to acquire a more compact intermediate faster (Figure 7), the folding assay does not seem to distinguish between properly folded and misfolded variants. As noted by the authors, formation of the folded form (NT) occurs at a similar rate, as does appearance of shed gp120 in the media. The authors concluded that co-translational cleavage of the M26P mutant results in rapid oxidation of gp160. However, it is not clear that intermediate steps in oxidation of gp160 are more rapid with the M26P signal sequence. Can this be quantified? The authors need to address this question in a revised version of the manuscript, perhaps commenting on whether there are other biochemical assays that might be able to distinguish between properly folded and misfolded variants. For example, are there anti-gp120 antisera that distinguish between unfolded and properly folded variants? If misfolded variants are being produced in the M26P mutant, are these variants subject to ERAD, or are these misfolded variants evading ER glycoprotein quality control? In spite of the tremendous efforts that this group and others have made to develop assays to detect steps in protein folding in a cell, one wonders whether the methods available to study this are still painfully insufficient and that even the cell's ability to monitor proper folding has its limitations.

2) What mechanism triggers eventual cleavage of the signal sequence of wild type gp160? It appears that the rate of folding (collapse of WT gp120 into the native form) occurs more rapidly than signal sequence cleavage (Figure 7). While we don't feel that the authors need to experimentally address this question in the current manuscript, the topic needs to be discussed in light of the current data. Do the fusion constructs that were assayed in Figure 2, Figure 4 and Figure 5 (gp160(1-89)-p2 of Lep, SP+10-GFP) undergo cleavage at rates similar to gp160? These experiments all used "steady-state" detection methods (e.g., Western blot), so it would appear that signal sequence cleavage is more strongly delayed or blocked in the fusion constructs. For example, in Figure 2, it is not clear why extending the gp160 linker residues appears to completely stop signal sequence cleavage under steady state conditions rather than just delaying it as it does in the context of gp160. Wouldn't you expect to primarily see the cleaved product under steady state conditions even if signal sequence cleavage was as slow as in gp120? The manuscript should be revised to clarify this point.

3) Figure 4 – the media should be shown to accurately conclude ER retention. It is not clear whether the cleaved protein is secreted or being degraded. And the protein bands should also be quantified to determine the retention levels of the cleaved compared to uncleaved protein.

4) For Figure 2, the microscopy could be moved to the supplementary materials as it is largely a control. SP+1 image shows two cells with apparently different patterns. Is this simply a difference in expression level of the SP+1 construct, or does it reflect different extents of ER retention?

[Editors' note: further revisions were requested prior to acceptance, as described below.]

Thank you for resubmitting your work entitled "Structure and topology around the cleavage site regulate post-translational cleavage of the HIV-1 gp160 signal peptide" for further consideration at *eLife*. Your revised article has been favorably evaluated by Ivan Dikic (Senior Editor), and a Reviewing Editor.

The manuscript has been improved but there are some remaining issues that need to be addressed before acceptance, as outlined below:

1) Figure 8. The text states that the mutant virus has caught up with wild type by day 6. The graph shows a data point for day 7, not day 6. Please correct.

2) Figure 8—figure supplement 1 is not mentioned in the Results section.

3) Figure 2 legend. The prolactin signal peptide (open box SP) and HIV signal peptide (grey box SP) both look like open boxes, at least on the PDF for review. The figure needs to have a darker grey in the HIV Signal peptide box.

4) Figure 6 and legend to Figure 6—figure supplement 1. The model for the wild type signal peptide looks like the green or red images from Figure 6—figure supplement 1, but the legend for the supplement states that the purple image from the supplemental figure was used for Figure 6. Please clarify.

5) Figure 6. There is a dotted blue line in this panel that isn't present in panels A-C.

6) Legend to Figure 4/4B. Figure 4—figure supplement 1 (# of cells analyzed) is not mentioned in the figure legend.

---

## [Author Response]

Essential revisions:

1) The authors would like to conclude that the altered folding kinetics of the M26P mutant yields a more heterogeneous population of gp160 proteins that would include misfolded variants, hence the less-fit phenotype. Unfortunately, the authors lack direct evidence that misfolded variants are produced. While the M26P mutant appears to acquire a more compact intermediate faster (Figure 7), the folding assay does not seem to distinguish between properly folded and misfolded variants. As noted by the authors, formation of the folded form (NT) occurs at a similar rate, as does appearance of shed gp120 in the media. The authors concluded that co-translational cleavage of the M26P mutant results in rapid oxidation of gp160. However, it is not clear that intermediate steps in oxidation of gp160 are more rapid with the M26P signal sequence. Can this be quantified? The authors need to address this question in a revised version of the manuscript.

We changed text to indicate more clearly that rather than a firm conclusion on an increase of the rate of oxidation or disulfide-bond formation in the M26P mutant, the data show that electrophoretic mobility had increased, which is indicative of increased compactness due to either more disulfide bonds or longer-distance disulfides or both. This mobility change can be quantified but this is inaccurate because of the prominent presence of untranslocated M26P protein on top of the folding-intermediate smear. Instead we ran M26P and wild-type gp120 next to each other and added lane profiles to show the large difference (Figure 7). Text was adapted accordingly. The limited-proteolysis assays we added (see below) confirm that long-distance disulfide bonds were formed in the M26P mutant.

Perhaps commenting on whether there are other biochemical assays that might be able to distinguish between properly folded and misfolded variants. For example, are there anti-gp120 antisera that distinguish between unfolded and properly folded variants?

Before submission of the manuscript we had analyzed already over a dozen conformation-specific antibodies without finding significant differences. Wild-type and M26P gp160 at later chase-time points shared many epitopes and we found no significant difference in the formation of functionally important epitopes in the CD4-binding site. We continued however with additional antibodies (unfortunately with decreased affinities) and now did find significant differences in recognition by antibodies directed against the inner domain of gp120 (around disulfide 54-74) and an epitope in gp41. Given the importance of the inner domain in triggering signal-peptide cleavage (discussed below) and its proximity to the signal peptide, it is fitting that early cleavage led to conformational defects in the inner domain. As this region interacts with gp41, this likely causes the defects we see in gp41. We now added these data as Figure 7 panels E and F with text.

An alternative folding assay we used with success involves proteolytic digestion under limiting conditions (low concentration protease, low temperature, short times). We added Figure 7 panels C and D and text. Limited proteolysis of wild-type and M26P gp120 at the 2-hour time point, when the majority of protein was “native”, did not yield a difference in proteolytic-fragment pattern (Figure 7). Early after synthesis however the difference was convincing and striking (Figure 7), demonstrating that in wild-type gp120 more local disulfide bonds formed, whereas in M26P all parts of gp120 were connected by disulfide bonds.

If misfolded variants are being produced in the M26P mutant, are these variants subject to ERAD, or are these misfolded variants evading ER glycoprotein quality control?

We have not seen any difference between wild-type and M26P gp120 or gp160 in terms of secretion/shedding, degradation or aggregation. Degradation of Env is not detectable; even after 24 hours of chase gp160 molecules still are folding and acquiring an ER-exit-competent conformation (Land et al., FASEB J 2003). Misfolding leads to aggregation rather than degradation. The M26P mutation may only seem to lower expression of gp120 or gp160 because targeting of this mutant to the ER is less efficient and part of Env remains untranslocated. We added more explicit text on this issue.

In spite of the tremendous efforts that this group and others have made to develop assays to detect steps in protein folding in a cell, one wonders whether the methods available to study this are still painfully insufficient and that even the cell's ability to monitor proper folding has its limitations.

Indeed, the many years of work we have done on protein folding in intact cells has provided a clear picture of the cell's frequent failure to recognize functional folding. This is underscored by diseases such as familial hypercholesterolemia or cystic fibrosis, where many patients suffer from mutant LDL receptor or CFTR, respectively that does leave the ER, does reach the cell surface, but is not functional.

Reviewers' statement that the methods may be painfully insufficient we took as a challenge, as we had recently added limited proteolysis to our panel of methods for disulfide-bond-containing proteins. Figure 7 shows the results. Nevertheless, misfolding indeed is usually much more subtle than commonly perceived.

2) What mechanism triggers eventual cleavage of the signal sequence of wild type gp160? It appears that the rate of folding (collapse of WT gp120 into the native form) occurs more rapidly than signal sequence cleavage (Figure 7). While we don't feel that the authors need to experimentally address this question in the current manuscript, the topic needs to be discussed in light of the current data.

Signal-peptide cleavage is not slower than the collapse. In Land et al. (FASEB J, 2003) we showed that cleavage can start at an electrophoretic mobility just above the native position.

We in the meantime have data to show that almost complete gp120 is needed for cleavage to occur. Shorter, C-terminally truncated constructs of gp120 do not lose their signal peptides. We predict that cleavage is triggered by integration of parts of the C1, C2 and C5 domains into a β sandwich in the inner domain. This integration likely triggers a conformational change in the N-terminus, changing the structure around the cleavage site from its predicted α helix to the β strand found in crystal structures of cleaved gp120. We now added text to discuss this.

Do the fusion constructs that were assayed in Figure 2, Figure 4 and Figure 5 (gp160(1-89)-p2 of Lep, SP+10-GFP) undergo cleavage at rates similar to gp160? These experiments all used "steady-state" detection methods (e.g., Western blot), so it would appear that signal sequence cleavage is more strongly delayed or blocked in the fusion constructs.

The Lep reporter construct was not analyzed in steady state but by radiolabeling in-vitro translations. We have clarified the text in the manuscript. The +10 GFP reporter construct did not undergo cleavage either, but in this case analysis was by Western. To address the question of whether only delayed or completely blocked we added a radioactive pulse-chase analysis on this construct, which shows that there is no post-translational cleavage anymore after the initial co-translational removal of the signal peptide (Figure 2—figure supplement 1).

For example, in Figure 2, it is not clear why extending the gp160 linker residues appears to completely stop signal sequence cleavage under steady state conditions rather than just delaying it as it does in the context of gp160. Wouldn't you expect to primarily see the cleaved product under steady state conditions even if signal sequence cleavage was as slow as in gp120? The manuscript should be revised to clarify this point.

Knowing what we know now, that we need almost complete gp120 to trigger cleavage, it is no surprise that neither the Lep nor the GFP constructs showed any post-translational signal-peptide cleavage.

3) Figure 4 – the media should be shown to accurately conclude ER retention. It is not clear whether the cleaved protein is secreted or being degraded. And the protein bands should also be quantified to determine the retention levels of the cleaved compared to uncleaved protein.

We have followed up on these experiments and discovered that our original interpretation was incorrect for an unexpected reason. We predicted that the ER-retrieval sequence, KDEL, was necessary to localize the signal-cleaved GFP reporter in the ER lumen, which we confirmed by fluorescence microscopy of the reporters with the KDEL sequence. The unexpected findings were of the controls, the reporters without KDEL sequences: the +5 GFP reporter lacking the KDEL retrieval motif turned out to be localized predominantly to the Golgi complex. This would be consistent with secretion of the cleaved reporter into the secretory pathway and exit from the ER. However, the immunoblot data indicate that the major pool in cells is the signal-uncleaved version. Thus, the Golgi complex-localized protein most likely still contains the signal peptide. Further consideration of the data suggests a model in which the no-KDEL variants are uncleaved not because of the structural accessibility of the signal cleavage site, but rather the segregation of the no-KDEL variants away from the ER and the signal-peptidase enzyme complex. This artifact renders the no-KDEL reporter variants irrelevant to our studies, because we know from previous studies as well as our own data in this study, that immature signal sequence-uncleaved gp120 and gp160 remain ER localized and only exit the ER upon signal cleavage.

These results in no way alter interpretation of the rest of our studies. They simply argue that the no-KDEL data should be removed to avoid any confusion. We adapted text accordingly.

4) For Figure 2, the microscopy could be moved to the supplementary materials as it is largely a control. SP+1 image shows two cells with apparently different patterns. Is this simply a difference in expression level of the SP+1 construct, or does it reflect different extents of ER retention?

Following the reviewer's request, Figure 2 has been moved to the supplementary material as Figure 2—figure supplement 1. We apologize for any confusion with the colocalization data. There is no difference in ER retention. The reporter and ER-localization reporters both perfectly colocalize. The difference in merge color is due to differences in transfection efficiency/expression levels of the co-transfected cells. The levels of the red ER-localization reporter is higher relative to the GFP reporter in the left cell, leading to a stronger red merge signal. We provided the individual channel images in grayscale to highlight the identical localization patterns, as well as the intensity/expression differences.

To avoid confusion, we have replaced the image for SP+1 with a single cell with comparable levels of expression of the ER-localization reporter and the GFP reporter.

[Editors' note: further revisions were requested prior to acceptance, as described below.]

The manuscript has been improved but there are some remaining issues that need to be addressed before acceptance, as outlined below:

1) Figure 8. The text states that the mutant virus has caught up with wild type by day 6. The graph shows a data point for day 7, not day 6. Please correct.

The manuscript has been corrected to show say day 7 instead of day 6.

2) Figure 8—figure supplement 1 is not mentioned in the Results section.

The manuscript has been updated to include reference to this figure.

3) Figure 2 legend. The prolactin signal peptide (open box SP) and HIV signal peptide (grey box SP) both look like open boxes, at least on the PDF for review. The figure needs to have a darker grey in the HIV Signal peptide box.

The figure has been updated to rectify this.

4) Figure 6 and legend to Figure 6—figure supplement 1. The model for the wild type signal peptide looks like the green or red images from Figure 6—figure supplement 1, but the legend for the supplement states that the purple image from the supplemental figure was used for Figure 6. Please clarify.

Our apologies for the confusion, upon review we noticed that there were several errors in the both the supplemental figure and the figure legend. We had intended to use the purple prediction in Figure 6 to align it with Figure 6 but had instead selected the wrong image. Furthermore, we erroneously included predictions for a different strain (Bg.505) of HIV in Figure 6—figure supplement 1. We have updated Figure 6—figure supplement 1 to show the predicted structure from the LAI signal peptide and included the correct image for Figure 6.

5) Figure 6. There is a dotted blue line in this panel that isn't present in panels A-C.

This has been removed in the updated version of the figure.

6) Legend to Figure 4/4B. Figure 4—figure supplement 1 (# of cells analyzed) is not mentioned in the figure legend.

We were told by the editorial team that we could not include a table as a supplementary figure during our resubmission. Instead the data has been placed as a source data file (Figure 4—source data 1). The legend has been updated to include reference to this. We have also updated the figure legend of Figure 7 for the same reason.